# Environmental Factors Affecting Wildfire Burned Area In South-Eastern France, 1970-2019

Christos Bountzouklis[1], Dennis M. Fox[1], Elena Di Bernardino[2]

[1]University of Côte d'Azur, UMR CNRS 7300 ESPACE, Nice, 06204, France

[2]University of Côte d'Azur, UMR CNRS 7351 LJAD, Nice, 06108, France

*Correspondence to*: Christos Bountzouklis (christos.bountzouklis@univ-cotedazur.fr)

**Abstract.** Forest fires burn an average of about 440,000 ha each year in southern Europe. These fires cause numerous casualties

and deaths and destroy houses and other infrastructures. In order to elaborate suitable fire-fighting strategies, complex interactions between human and environmental factors must be taken into account. In this study, we investigated the spatio-temporal evolution in burned area over a 50-year period (1970-2019) and its interactions between topography (slope inclination and aspect) and vegetation type in south-eastern France by exploiting Geographic Information System databases. Burned area decreased sharply after 1994, with the advent of the new fire suppression policy which focused on rapid extinction of fires in

their early phase. The geographic distribution of burned area has also changed in the last 25 years, mainly in regions where large fires occurred (Var department). In other parts, even though forest fires are still frequent and occur in the same geographic locations, the total extent of the burned area is significantly reduced. Slope orientation presents an increasingly important role every decade, S-facing slopes have the greatest burned areas, while the opposite is observed for N-facing and W-facing ones. Fire increasingly favors intermediate slopes after the sharp decrease of burned area in 1990. The largest part of the BA is

strongly associated with the location of sclerophyllous vegetation clusters, which exhibit high fire proneness while simultaneously expanding the region.

## 1 Introduction

Forest fire is a common and important element of the earth system (Bond and Keeley, 2005) that is capable of severely disturbing natural ecosystems and threaten human welfare and wellbeing throughout the globe. The Mediterranean climate is

characterized by hot and dry summers which favor fire ignition and propagation. Consequently, wildfires are particularly active around the Mediterranean basin, and Mediterranean-climate areas are considered to have a wide range of environmental and socioeconomic impacts (Miller et al., 2009; San-Miguel-Ayanz et al., 2013; Ganteaume et al., 2013).

Forest fires burn an average of 440,000 ha each year in the Euro-Mediterranean region which corresponds to about 85% of the total burned area (BA) in Europe (San-Miguel-Ayanz et al., 2020). Of the 5 principal Euro-Mediterranean countries concerned

by forest fires (France, Greece, Italy, Portugal and Spain), France has the lowest amount of BA (San-Miguel-Ayanz et al.,

2020). It also has the smallest potential burnable area since only the southern Mediterranean fringe is concerned by forest fires. France, Spain, Italy, and Greece all show similar trends in decreasing decadal BA in 1980-2010, and only Portugal has experienced a progressive increase during this interval (San-Miguel-Ayanz et al., 2020). It should be noted that BA are generally decreasing despite increases in summer temperatures throughout the Euro-Mediterranean zone (Pokorná et al., 2018;
Rodrigues et al., 2020), and this can be attributed to more efficient fire-fighting strategies (Fox et al., 2015; Turco et al., 2016; Ganteaume and Barbero, 2019).

Forest fire spatial distribution, size, and frequency are associated with several factors that are highly interactive and can be categorized into two main groups: i) environmental and ii) anthropogenic. Environmental factors generally include fuel characteristics (e.g type, water content), topography (e.g slope inclination, altitude, aspect) and weather conditions (e.g
temperature, humidity, wind speed); anthropogenic factors include the characteristics of the transitional zone between wildland vegetation and artificial areas in the Wildland Urban Interface (WUI).

Among the environmental characteristics, several studies provide evidence of spatial patterns relating topography to forest fire probability (Dickson et al., 2006; Padilla and Vega-García, 2011) and burnt area (Nunes et al., 2016). Slope aspect affects incoming solar radiation and can determine the fuel type, fuel moisture, and fuel density which all influence flammability
(Holden et al., 2009). In addition, aspect influences the degree of ecological change related to fire (fire severity) (Birch et al., 2015; Estes et al., 2017; Parks et al., 2018). In the northern hemisphere, south-facing slopes receive more solar radiation during the day than north-facing slopes, and this can enhance burn severity (Alexander et al., 2006; Oliveira et al., 2014a; Oliveras et al., 2009) but the trend is not systematic (Broncano and Retana, 2004). In addition to the impact on fire severity, other studies (Mouillot et al., 2003) have demonstrated that south-facing slopes in Corsica (France) can burn more frequently than other
exposures. On the North shore of the Mediterranean, South-facing slopes frequently have more housing than north-facing slopes, and this may contribute to a greater number of ignitions (Fox et al., 2018). Steep slopes have higher rate of spread and fire intensity (Capra et al., 2018), as well as increased fatality rate over flat areas (Molina-Terrén et al., 2019). Csontos and Cseresnyés (2015) observed an exponential increase in upslope fire spread with the increase in slope inclination whereas downslope fire spread velocity was unaffected by slope angle and was similar to rates detected on flat terrain. Slope and altitude
tend to be correlated but their association with fires is often conflicting. For instance, Nunes et al., (2016) studied BA and ignition density on a municipal scale in Portugal and found both are positively affected by elevation and slope. Similarly in Elia et al., (2019) results showed that the probability of fire ignition increased with elevation and slope in southern Italy. However, other studies such as Narayanaraj and Wimberly, (2012) observed that elevation and slope had a negative association with human-caused fires. The role of vegetation is complex and can be influenced by flammability (Michelaki et al., 2020;
Molina et al., 2017) or spatial patterns of vegetation in the landscape (Curt et al., 2013). Vegetation continuity affects fire propagation which contributes to determine BA (Duane et al., 2015; Fernandes et al., 2016). Vegetation type is another important factor to consider which has explored in number of studies though fire selectivity indices (Bajocco and Ricotta, 2008; Barros and Pereira, 2014; Carmo et al., 2011; Moreira et al., 2009; Moreno et al., 2011; Nunes et al., 2005; Pereira et al., 2014). Overall, there is a widespread agreement in literature that shrublands are regarded as fire prone areas at multiple

scales: regional (Carmo et al., 2011; Moreno et al., 2011), national (Nunes et al., 2016, 2005) and continental (Moreira et al., 2011; Oliveira et al., 2014b; Pereira et al., 2014) scale.The probability of large fires is greater in dense shrublands than in forested ecosystems in the Mediterranean basin (Moreira et al., 2011; Ruffault and Mouillot, 2017). According to Mermoz et al., (2005) fire proneness of shrublands could be related to their recovery rate since shrublands can regenerate faster and favor fuel accumulation in a short time unlike forests which take longer to recover and spread. In addition,Oehler et al., (2012) point

out that shrubs are considered as a low suppressing priority by fire fighters due to the low cost of its restoration. Other vegetation types, such as grasslands, are also considered to be fire prone in Europe (Oliveira et al., 2014a). Cultivated areas are the least fire prone types mainly because of their low combustibility and their geographic proximity to built-up land covers which facilitates rapid fire detection and suppression (Moreira et al., 2011). Forested areas are found to be more fire prone than cultivated areas but less than shrublands (Moreira et al., 2011). More specifically, broadleaved forests are usually less

prone to burning than coniferous species which present a greater fire hazard (Moreira et al., 2009; Oliveira et al., 2014a). Several of the aforementioned factors do not remain constant in the spatial and temporal domain and thus determining the relative influence of changes in biomes, climate, but also in fire management practices is crucial both for policy-making and fire management (Bowman et al., 2017). There are numerous recent efforts that aim to analyze spatial and temporal trends of fire activity at a global, national and regional level. Otón et al., (2021) analyzed global trends of BA based on the FireCCILT11

database which is the longest available global BA dataset to date (1982-2018). At a national level Catarino et al., (2020) investigated the trends of annual BA in Angola between 2001 and 2019 using MODIS products (MCD64A1) and associated the significant trends to land cover, ecological regions and protected areas. Ganteaume and Barbero, (2019) utilized a long-term (1957-2017) fire geodatabase to analyze spatio-temporal variations of large fires in terms of frequency and BA, in the French Mediterranean. Silva et al., (2019) used a satellite derived BA dataset covering a 39-year period over the Iberian

Peninsula to study BA trends and explore the relationship between areas with significant BA trends and fire danger. Urbieta et al., (2019) studied the spatio-temporal trends in Spain between 1980 to 2013 with regard to fire frequency, BA and fire size, and their relationship with changes in climate, land-use and land-cover, and fire suppression. Viedma et al., (2018) assessed the changing role of environmental and human-related factors in reference to fire activity, in west-central Spain from 1979 to 2008.Fire suppression is also among the factors that can influence fire spread. In France, as a response to the large fires that

occurred between 1986 to 1990 a major change in fire suppression strategy was established in the 1990s, that focused on rapid suppression of fire ignitions regardless of the weather conditions in order to avoid the propagation of fire (DSG ref). The fire policy had a significant impact in fire activity in fire-prone areas like in Southern France and weakened the fire-weather relationship (Ruffault and Mouillot, 2015). Despite the sharp decrease in BA after the full implementation of the fire management policy, its efficiency on very large fires is not as successful as for smaller fires, since changes in BA that

correspond to large return periods don't seem significant (Evin et al., 2018). Many studies have focused on determining relationships between fire behavior and driving factors (Mhawej et al., 2015), and the relative level of importance of factors can vary from one region to another depending on the environmental and socioeconomic contexts and the scale of a study (Moritz et al., 2005; Lafortezza et al., 2013; Ganteaume and Long-Fournel,

2015). Few studies have examined how efficient fire suppression strategies impact the spatial distribution of BA. Identifying spatial patterns and the main driving forces that determine fire distribution provides useful information for fire and civil protection agencies, and it assists in allocating appropriate firefighting resources and in designing proper prevention actions especially in the Mediterranean area (Moreira et al., 2011).

The objective of this study is to quantify changes in BA spatial and temporal patterns induced by a major shift in fire suppression strategy that was initiated in the early 1990s in South-eastern France. The time interval under study spans 5 decades (1970-2019) and includes the relation of BA with respect to environmental factors such as a) topography (slope inclination, slope orientation and b) vegetation type. Although several studies have investigated the relationships between BA and environmental factors, very few have covered such a long-time interval based on burn scar polygons, nor have they been explicitly related to changes in fire suppression methods.

## 2 Data and Methods

### 2.1 Study area

The study area is comprised of a subset of the 3 administrative departments with the greatest BA in continental France (only Corsica has greater burned area) according to the official database for forest fires in the French Mediterranean area (promethee.com): Bouches-du-Rhône, Var, and Alpes-Maritimes (Table 1, Fig. 1). The areas within the departmental limits that were excluded, represent surfaces that cannot burn such as: i) marshlands in the westernmost part of Bouches-du-Rhône and ii) high alpine mineral surfaces located in the northern part of Alpes-Maritimes.

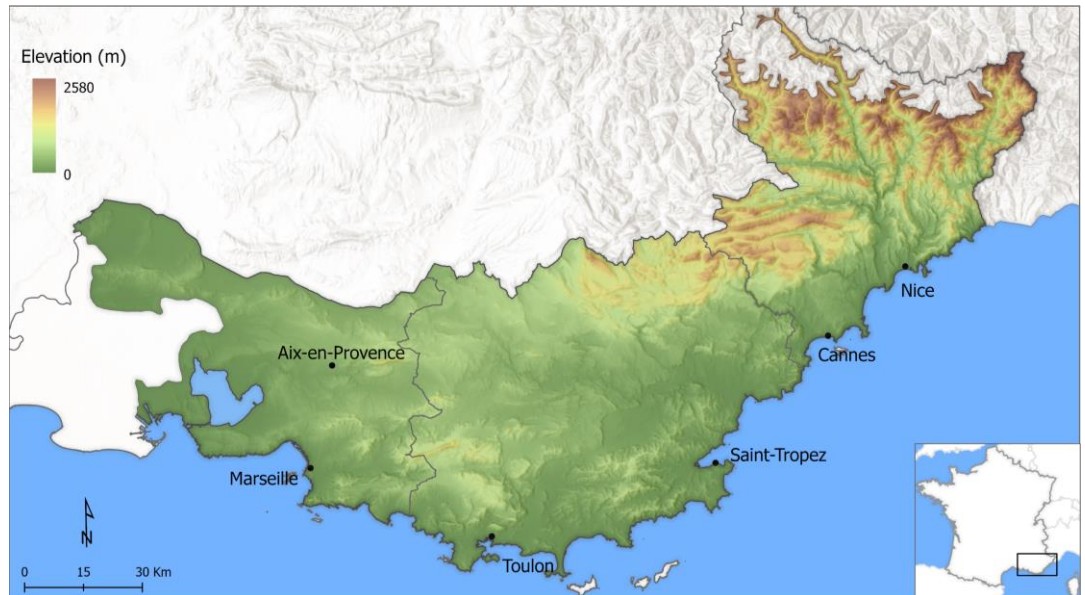

**Figure 1: Map of south-eastern France showing the study area and the departmental limits overlaid on a 5 m Digital Elevation Model.**

Table 1: Environmental characteristics of the study area per departmental unit

|  | **Bouches-du-Rhônes** | **Var** | **Alpes-Maritimes** |
|---|---|---|---|
| **Total area (km²)** | 3456 | 6019 | 3495 |
| **Forested area (km²)** | 1530 | 4044 | 2727 |
| **Ratio forest/total (km²)** | 0.44 | 0.67 | 0.78 |
| **Mean slope (°)** | 8.8 | 11.9 | 24.3 |
| **Median slope (°)** | 5.7 | 9.6 | 25.2 |

Topography varies noticeably from west to east (Fig. 1). The gentlest slopes are found in the west (Bouches-du-Rhône) and both altitude and inclination increase towards the east. The steepest slope inclinations are found in the northeastern part of the study area where the French Alps are located. Topography influences population distribution since much of the built area is concentrated along the coast or on shallow to intermediate slopes in the WUI. In the Bouches-du-Rhône, the western portion of the department has particularly low population densities due to the presence of the national park and wetlands mentioned above. Similarly, much of the population in the Alpes-Maritimes is concentrated in the southern portion of the department. The 2010 population densities of 388.8, 167.5, and 252.0 persons/km² for the Bouches-du-Rhône, Var, and Alpes-Maritimes, respectively, are therefore only gross approximations as they simply divide population by total area without accounting for geographic distributions. The order, however, is accurate and shows the greatest population density for Bouche-du-Rhône, and the lowest for the Var. Based on the demographic and environmental characteristics described above, the westernmost section (Bouche-du-Rhône) of the study area has low potential for fire ignition and propagation but increases when moving towards the eastern half of department. The central part of the study area (Var department) has a high potential for fire ignition and the greatest potential for fire propagation since it has a high forested area and a large continuous WUI area. Finally, the eastern section (Alpes-Maritimes department) has high ignition and propagation potentials in the southern portion of the department and low ignition / high propagation at higher altitudes.

## 2.2 Fire database

Forest fire research in France is usually based on the national database for forest fires in France (www.promethee.com) where fire location is described by municipality where ignition occured. For this study, we used a fire Geographic Information Systems (GIS) database provided by the National Forestry Office (Office National des Forêts, ONF) and the Delegation for the Protection of the Mediterranean Forest (Délégation à la Protection de la Forêt Méditerranéenne, DPFM). Even though the number of recorded fires is significantly lower than the Promethee database, the total area burned in a given time is almost identical. To the best of our knowledge, this is only the second exploitation of this geodatabase after Ganteaume and Barbero (2019). The dataset includes more than 3,000 digitized burn scar polygons for fires that occurred between 1970 and 2019. Due

to the long temporal extent of the database, the accuracy and the methods used to define burn scars varied over time. In the 1970s, burn scars were mapped using field measurements with GPS devices, and the technique progressively evolved to integrate remote sensing data (satellite imagery, orthophotos). Although the description of how BA was defined is not recorded in the database, earlier polygons are clearly less accurate (coarse shapes with little detail) than burn scars after the advent of satellite imagery (Fig. 2).

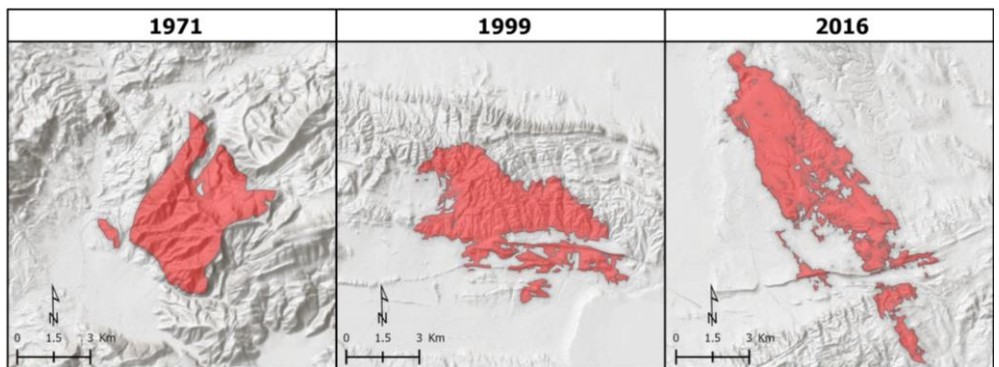


**Figure 2: Evolution of digizited burn scar accuracy over the past decades.**

### 2.3 Environmental variables

### 2.3.1 Topography

Burn scar polygons were rasterized to a 5 m spatial resolution and overlain on a 5 m Digital Elevation Model (DEM) extracted
from RGE-ALTI©, the official National Geographic Institute (Institut Géographique National, IGN) database. The DEM was used to calculate aspect and slope inclination (Fig. 5). In the conversion of vector polygons to raster cells, BA polygons smaller than half the cell size (25 m²) were not defined as burned during rasterization, so BA for the aspect and slope inclination analyses represent approximately 96 % of actual BA in the study area.. Aspect was divided into 5 categories: flat, north, east, south and west.. Slope was divided into 5 categories: 0-10°, 10-20°, 20-30°, 30-40° and>40°.

### 2.3.2 Vegetation type

For the computation of the forested BA and the identification of fire-prone vegetation categories, GIS forest layers were extracted from the European CORINE land cover (CLC) database. The database includes five reference years 1990, 2000, 2006, 2012 and 2018. In addition to the CLC reference layers, it was considered best to backcast two additional forest cover layers for 1972 and 1980 to account for any transitions between forested and non-forested surfaces for the two decades
preceding the CLC database. The methodology followed for the projection process is addressed in Subsection 2.5.1. The fire geodatabase was then matched with the CLC layer that was chronologically closest to the equivalent fire period (see Table 2).

**Table 2: Corine land cover layers and their respective fire periods.**

| Corine Land Cover | Fire period |
| --- | --- |
| 1972 (Projected) | 1970 – 1974 |
| 1980 (Projected) | 1975 – 1984 |
| 1990 | 1985 – 1994 |
| 2000 | 1995 – 2002 |
| 2006 | 2003 – 2009 |
| 2012 | 2010 – 2014 |
| 2018 | 2015 – 2019 |

The vegetation types that were used in the current study follow the CLC nomenclature: Broad-Leaved Forest, Coniferous Forest, Mixed Forest, Natural Grassland and Sclerophyllous Vegetation (Fig. 4). Although Natural grassland and Sclerophyllous vegetation are not forests, the categories will be referred to collectively as wildland or forested areas indiscriminately for the sake of brevity.

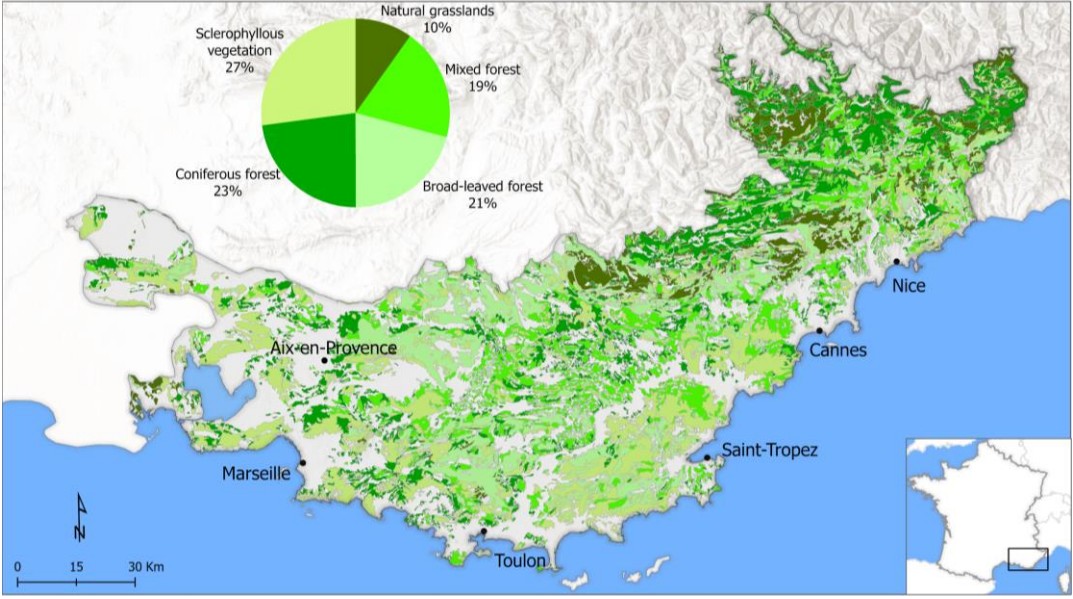

**Figure 3: Distribution of vegetation types based on CLC 2018.**

### 2.3.2.1 Forest layer projection

Although most urban growth occurred on agricultural land (Roy et al., 2015) and forest cover changed little, the Land Change Modeler (LCM) module of Terrset (Eastman 2020) was used to predict vegetation cover in 1972 and 1980. LCM is programmed to forecast change from an earlier to a later date, so going back in time required the temporal inversion of

filenames for the 1990 (renamed to 2000) and 2000 (renamed to 1990) CLC layers; in this way, land cover was simulated for 2010 (1980) and 2018 (1972). Land cover categories were simplified from the original CLC categories to the following: Built, Broad-leaved Forest (Broad), Coniferous Forest (Conifer), Mixed Forest, Natural Grassland (Grass), Sclerophyllous Vegetation (Bush), Other, and Water. Only transitions greater than 0.05 % of the landscape (14.3 km²) were modeled, and these included the following (From-To): Bush-Grass, Bush-Other, Built-Other, Grass-Other, Broad-Bush, Other-Grass, Bush-

Conifer, Other-Bush, Bush-Broad, Bush-Mixed, Mixed-Bush, Other-Conifer, Mixed-Broad, Mixed-Other, Other-Broad, Other-Mixed, Broad-Other, Grass-Bush, Mixed-Conifer, Built-Mixed, Built-Bush, Conifer-Mixed. Note that these are the inverse of historical trends, so the Built-Mixed transition actually backcasts the actual historical transition of Mixed Forest to Built area. Explanatory variables used to predict land cover change were the following: Altitude, Slope inclination, Distance from Built area, Distance from Broad, Distance from Conifer, Distance from Mixed, Distance from Grass, Distance from Bush,

Distance from Other and Distance from Water. According to Eastman (2020), Cramer's V values of ≥0.15 for explanatory variables are useful and should be kept in the model, and all explanatory variables used here met this criterion. Accuracy rates to model transitions ranged from 65 % to 90 % with mean and median values of 78 and 80 %, respectively.

**2.4 Spatiotemporal analysis of burned area 1970-2019**

    A 500x500 m grid (25 ha) was created and overlaid on the study area in order to measure the percentage of each 25 ha cell

that was burned each year between 1970 and 2019 (50 years) (Fig. 4). These percentage values were then summed to produce the cumulative percentage of BA for each cell. This approach facilitated the effort to identify clusters of cells/areas that have been burned multiple times through time. To better illustrate the spatial and temporal trends in the study area, the aforementioned methodology was applied to two 25-year subsets of the fire dataset i) 1970-1994, and ii) 1995-2019. The distinction between the earlier and later intervals was defined to highlight the impact of suppression strategies on fire

occurrence as the break corresponds roughly to a major shift in firefighting strategy and allocated resources in France.


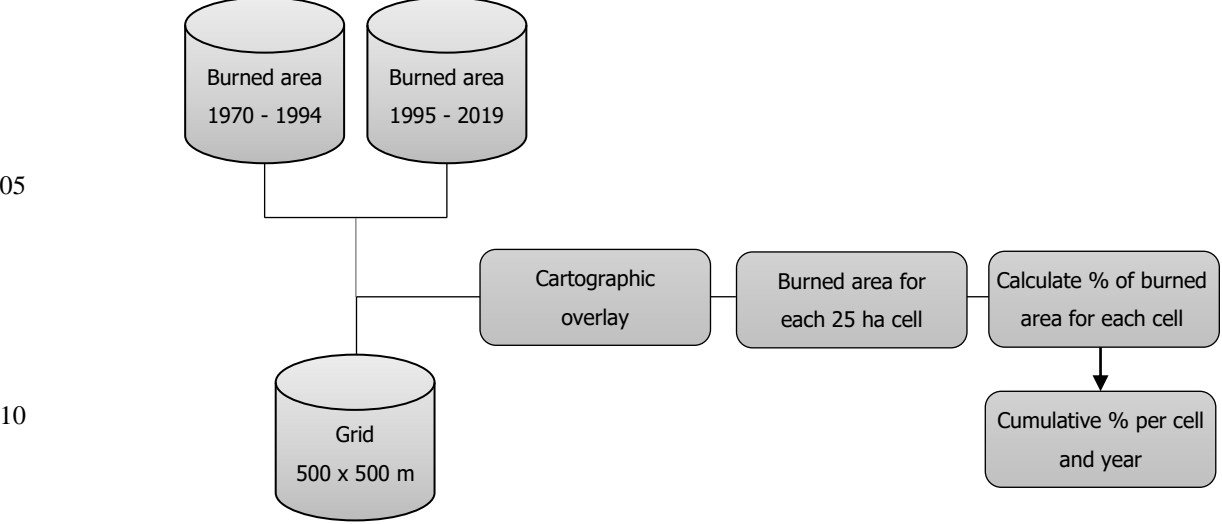


**Figure 4: Flow chart depicting the processing steps to generate the cumulative percentage of forested burned area per cell.**

In order to identify spatiotemporal trends within the entire time period (1970-2019), a modified version of the Mann-Kendall test was applied (Kendall 1975; Mann, 1945). The Mann-Kendal test is a non-parametric test which is used to statistically assess monotonic upward or downward trends for a variable through time. In this study we used the contextual Mann-Kendall (CMK) test which was introduced by Neeti and Eastman (2011) and it differs from the original one since it evaluates trends at a 3x3 cell neighbourhood for each cell in a grid. The CMK method has been used to assess trends in BA with satisfactory
outcomes (Silva et al., 2019; Catarino et al., 2020; Otón et al., 2021).

The specific method was established on Tobler's First Law of Geography (Tobler, 1970) which states that "everything is related to everything else, but near things are more related than distant things". By assuming that trends show signs of spatial autocorrelation between adjacent cells, the method allows for greater confidence in identifying the presence of a trend (Neeti and Eastman, 2011). However, the test requires observations to be a set of independent random variables and thus applying
the test on data that are temporally autocorrelated may lead to false rejection of the null hypothesis of no trend (Douglas et al., 2000). To assess the temporal autocorrelation in our dataset we applied the Durbin-Watson test (Durbin and Watson, 1950), and to remove it, the prewhitening procedure by Want and Swail, (Wang and Swail, 2001) which preservers the same trend but without the temporal autocorrelation (Fig. 5).


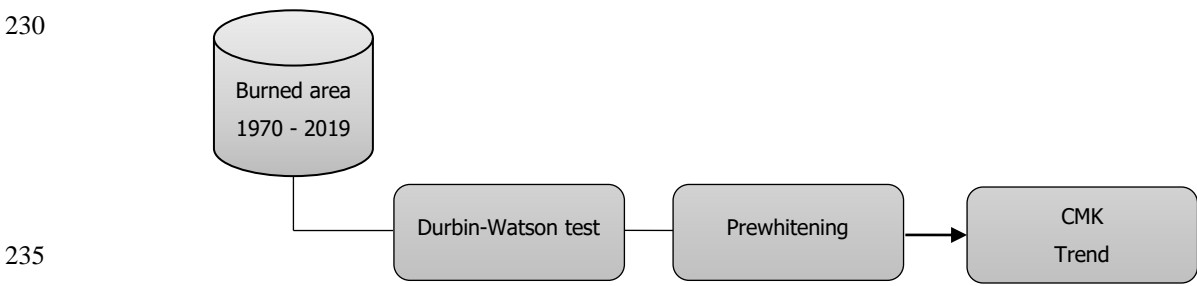

**Figure 5: Flow chart depicting the processing steps to estimate trend significance using the Contextual Mann-Kendall method.**

**2.5 Fire Selectivity (Jacobs' Index)**

In order to examine the fire proneness of the environmental variables considered in this study (slope, aspect, vegetation type) between the two 25-year periods we calculated a resource selection index. Even though resource selection is based primarily on wildlife ecology (Manly et al., 2002), there are several studies that applied similar methods for fire selectivity (Bajocco and Ricotta, 2008; Barros and Pereira, 2014; Moreira et al., 2001, 2009; Moreno et al., 2011; Nunes et al., 2005; Oliveira et al.,
2014a). The rationale behind fire selectivity is that fire burns selectively when the proportion of a class (e.g a type of vegetation)

within a burned area is higher than the proportion of the available area to burn. The opposite applies If a specific class of variable is burned proportionally less than the proportion available within an area (fire avoidance).

In our work, we used Jacob's selectivity index (Jacobs, 1974) which is defined as:

$$D_i = \frac{r - p}{r + p - 2rp} \quad (1)$$

**R** stands for the proportion of a resource class **i** used by fire and **p** is the proportion of a resource class **i** available to fire. Jacobs' index values range between -1 and 1. Positive values indicate fire preference while the negative ones indicate fire avoidance. The index was calculated for each class of the environmental factors (described in the subsequent sections) for each year. Considering that the area under is relatively small, the available area to burn is defined as the total forested area in the region (Fig. 6)

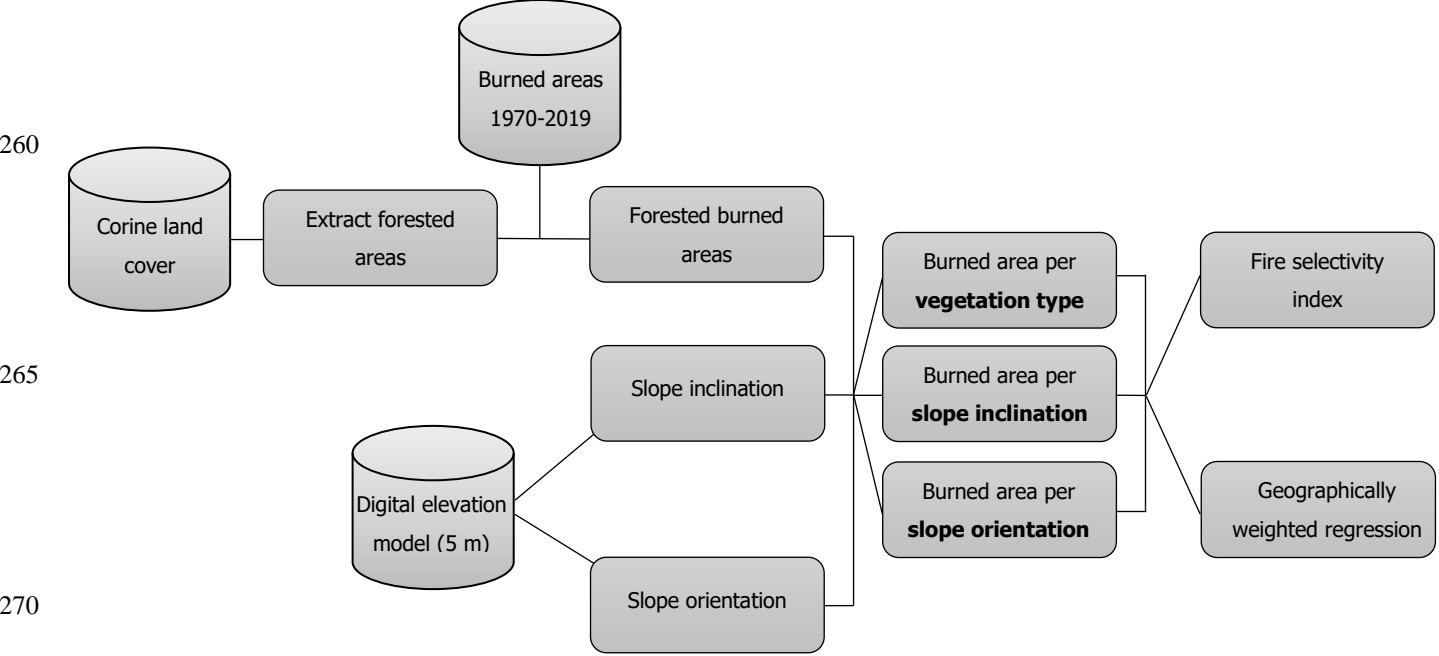

**Figure 6: Flow chart depicting the processing steps and data used to relate BA to vegetation type, slope orientation and inclination.**

**2.6 Geographically weighted regression**

To evaluate the importance of the environmental factors affecting BA and to observe whether their importance was impacted by the shift in fire suppression, a geographically weighted regression (GWR) was used. Applications of GWR can be found in variety of interdisciplinary fields including forest fires (Koutsias et al., 2010; Martínez-Fernández et al., 2013; Nunes et al., 2016; Rodrigues et al., 2016; Kolanek and Szymanowski, 2021). GWR is a local non-parametric regression method

(Fotheringham et al., 2003) that allows the relationships between dependent and explanatory variables to vary over space. The basic form of a GWR model, provided by Fotheringham et al. (1998, 2003) is defined as:

$$y_{i=\beta_{i0}} + \sum_{z=1}^{j} \beta_{ik} \, x_{ik} + \varepsilon_i$$

Where $y_i$ is the dependent variable at location $I$, $x_{iz}$ represents the $z$th explanatory variable at location $I$; $j$ is the number of explanatory variables, $\beta_{i0}$ is the intercept parameter at location $I$; $b_{iz}$ is the local regression coefficient for the $z$th explanatory variable at location $I$ and $e_i$ denotes the random error at location $i$. Since GWR allows coefficients to be spatially heterogeneous, a sub-model for the location of each observation is created that considers only a subsample of the total observations, where observations in closer proximity have higher effect in determining the local set of coefficients than observations located in larger distance (Fotheringham et al. 1998). This neighbourhood is called "kernel", while the maximum distance from a regression point at a location $I$ is defined as "bandwidth". The bandwidth is an important parameter than can be defined in two different ways: i) fixed bandwidth, (fixed distance for each regression point) and ii) adaptive bandwidth (fixed number of nearest neighbours for each regression point). The first type of neighbourhood is more appropriate when data are regularly distributed across apace whereas the second type is more appropriate for data that form spatial clusters. In the current work the adaptive bandwidth approach was utilized to fit the GWR model optimized based on the value of Akaike Information Criterion (Akaike, 1998). For each environmental variable mentioned in subsection 2.3.1 and 2.3.2, a univariate GWR model was used to explore the relationship with the dependant variable (% of BA) for two 25-year periods i)1970-1994 and ii)1995-2019.

## 3. Results

Results presented below will first describe fire history for the 1970-2019 interval and then analyze the spatio-temporal evolution of BA split according to the two 25-year periods. Finally, it will explore the relationship of BA to topography (slope orientation and inclination) and vegetation type. Factor-specific results will be discussed as they are presented in the following results sections while broader considerations will be explained in the Discussion section.

### 3.1 Fire history 1970-2019

In total, 3,382 fires burned 296,820 ha in 1970-2019. The mean and mediean areas of BA are 87.7 ha and 4.2 ha, respectively; these values reflect the typical highly negatively skewed distributions of fire size where the vast majority of fires are small. The number of fires equal to or greater than 100 ha, 500 ha, and 1,000 ha is 378 (11.2%), 123 (3.6 %) and 65 (1.9 %), respectively. Of the total number of fires, 2,424 (88.2 %) occurred in forested landscapes, and these burned an area of 263,645 ha (88.8 % of total BA).

. Mean and median values for forested landscape fires area are slightly greater than for all fires at 111.7 ha and 6.5 ha, respectively. The number of fires equal to or greater than 100 ha, 500 ha, and 1,000 ha is 314 (13.0 %), 106 (4.4 %), and 60

(2.5 %), respectively. As stated above, results presented below will deal exclusively with the forested BA that was occupied by one of the vegetation types mentioned in section 2.4.2 since the trends with respect to vegetation and topography for all
fires and forested landscapes are nearly identical.

Annual forested BA varies significantly from year to year (Fig. 7) although there are clear differences between the first two decades (1970-1990) and the last three (1991-2019). Most of the big fires occurred in the 1980s followed by a sharp decrease in the early 1990s. Similarly to the rest of the southern Mediterranean Europe, the majority of the forested BA is related to a small number of large fires (Turco et al., 2016). Only 5 years (1979, 1986, 1989, 1990 and 2003) of the 50-year record account
for almost half of the total forested BA (126,700 ha). The forested BA for each of these years surpasses 20,000 ha, attaining nearly 36,000 ha in 1989. Of the 5 years cited above, only 2003 is found in the second 25-year interval. As described by Fox et al., (2015) for the Alpes-Maritimes, this corresponds to an improvement in fire-fighting strategy, and the same explanation appears to hold for the neighboring departments.

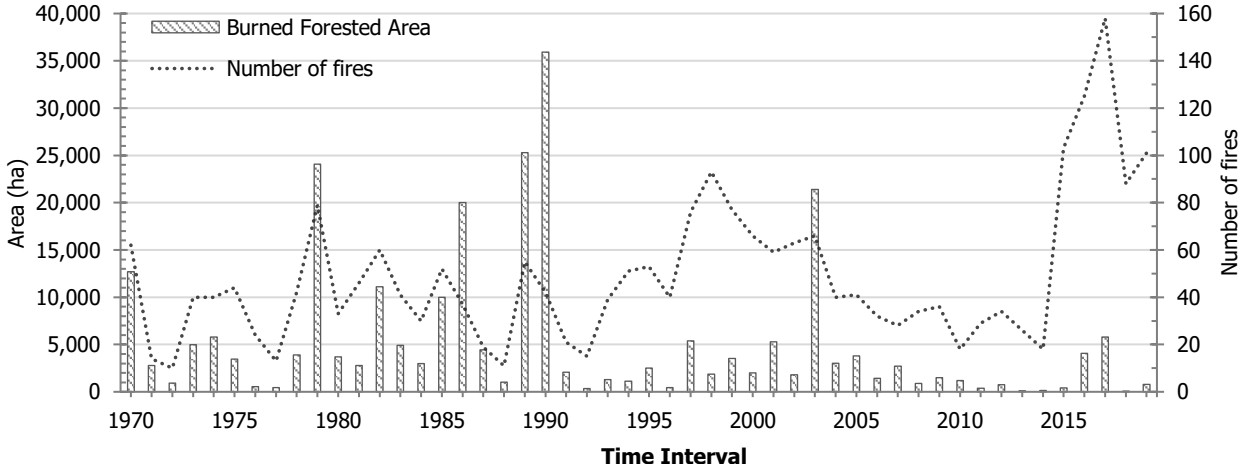

**Figure 7: History of annual forested burned area and number of fires from 1970 to 2019.**

### 3.1.1 Spatiotemporal analysis

Figure 8 maps cumulative percentage area burned inside each 25 ha cell for 1970-1994 and 1995-2019. Generally, most fires occur in the WUI north of the large coastal cities since densely developed areas have too little vegetation to burn, and relatively remote areas have too few ignition sources. Although we did not treat wind direction or speed, BA shapes in both periods tend
to align themselves with known wind patterns in the region: they have a NW-SE orientation throughout most of the western and central sections (Bouches-du-Rhône and Var departments) but show little preferential orientation in the eastern department of Alpes-Maritimes where wind speeds are lower than the "Mistral" winds in the Rhône valley. There is a clear difference between the two periods with the second one having significantly fewer burned cells, which are also slightly more spatially dispersed. In addition, cumulative percentage values are noticeably lower with a small number of cells exceeding 100 % and

only a few reaching 300 %. All major hotspots disappear in the second interval apart from some located mainly in the western area of study zone near Aix-Marseille.

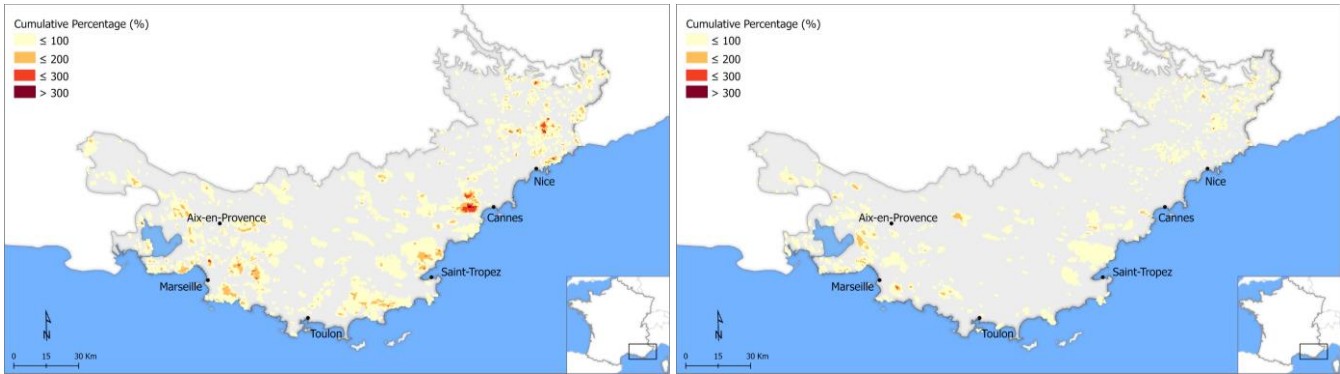

Figure 8: Cumulative percentage of forested burned areas in the 1970-1994 interval (left) and in the 1995-2019 interval (right) over a 500 x 500m grid.The largest patches in both intervals are found in the central part of the study zone which combines

continuous forest cover and a lower population density that is distributed more evenly throughout the Var department. Additionally, the two largest continuous BA clusters are found here, one north of Saint-Tropez and one east of Toulon (near Hyères). In the second time interval, the first cluster shrunk whereas the second one completely disappeared. In the western section of the study area, burned patches are located in constrained areas between densely built zones (Aix-en Provence and Marseille) with several cells displaying high fire recurrence. . In the eastern section of the study area, where population is

particularly dense along the coast (Cannes-Nice), BA cells are concentrated inland along the periphery of the coastal built-up area. A major hotspot with the highest cumulative percentage burned area is found just west of Cannes, and this patch almost disappears in the second period. In comparison to the rest of the study area, patches in the eastern department of the Alpes-Maritimes are smaller and more numerous with high to very high recurrence, even at high altitudes . A noteworthy difference between the first and the second intervals in both the western and the eastern segments of the study zone is that even though

the number of burned cells appears to be similar, the clusters of cells are smaller, which may indicate that fire ignitions were contained more quickly in the second interval.

Results of the CMK method depict areas of increasing and decreasing trends in terms of mean annual BA over the study area (Fig. 9). Positive Z-scores (colored in red) correspond to areas with increasing trends and negative Z-scores (colored in blue) correspond to areas with decreasing trends. Overall, a general decreasing trend of BA throughout most of the study area can

be observed, with approximately 60% of the cells corresponding to a negative value. The largest clusters of negative Z-scores are located predominately in the central areas of the region like W of Cannes (major hotspot that disappears in the second period), N of Saint-Tropez, N of Toulon and NE of Marseille). Clusters of positive Z-scores are much more constrained in terms of size and are generally dispersed. Significant decreasing trends are relatively limited and can be spotted in areas like N of Nice, W of Cannes and E of Marseille. On the contrary, significant positive trends are detected in several locations

(although limited in area) such as between Aix-en-Provence and Marseille and in the northeastern part (Alpes-Maritimes

department) of the study area. Zones near the biggest cities in the region (Nice, Marseille and Aix-en-Provence), present a contrasting pattern of mixed increasing/decreasing trends often at very short distances from one another.

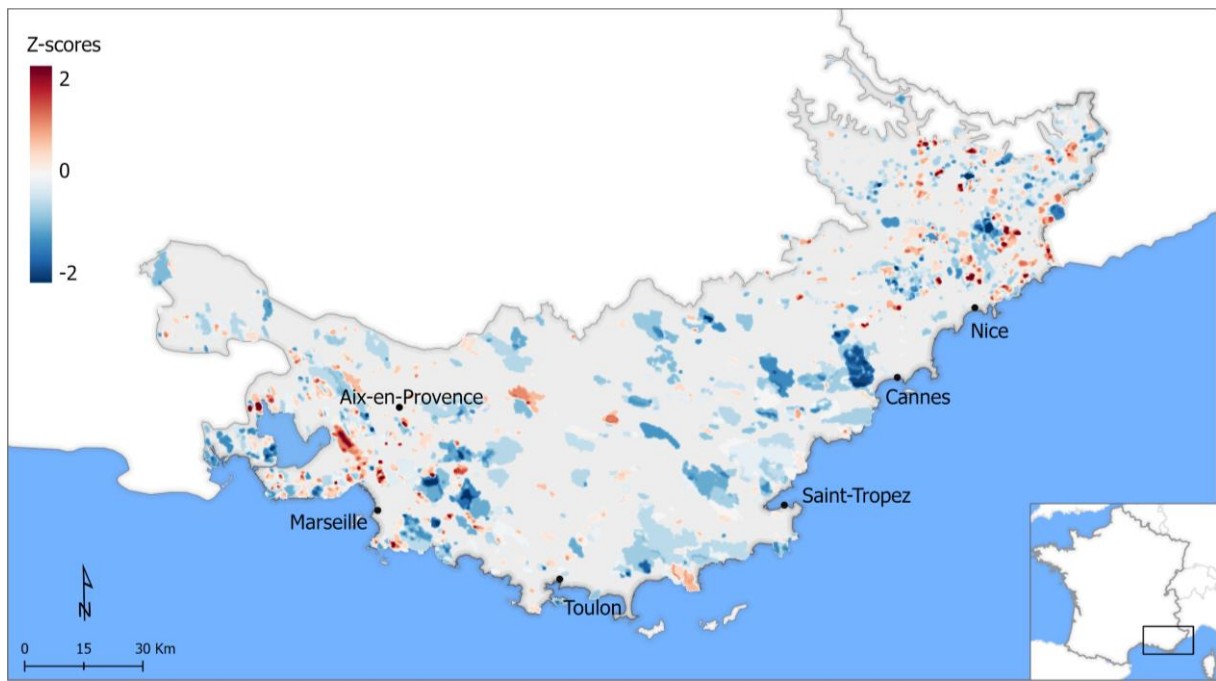

**Figure 9 Trends of mean annual burned area between 1970 to 2019 based on the Contextual Mann-Kendall method. Areas with positive Z-scores depict increasing trends of burned area, while negative Z-scores show decreasing trends.**

### 3.2 Fire selectivity and topography

Topographic effects studied here include bothaspect,andslope inclination. Since some areas may have greater BA values simply because in a given topographic class is more frequent in the landscape, Jacob's selectivity index was calculated in order to identify potential classes of slope aspects and slope inclinations that are preferred by fire between two periods: i)1970-1994 and ii)1995-2019.

### 3.2.1 Slope orientation

Figure 10 shows fire preference (Jacobs' index >0) and fire avoidance (Jacobs' index <0) for the two 25-year periods under study. Between 1970-1994, S-facing slopes present the highest median value (0.07) followed by W (0.04) and E (0.02) facing slopes. Flat areas have the lowest median value (-0.29)  and N (-0.15). In the second period (1995-2019), the median fire selectivity of S-facing slopes (0.26) is increased and presents a clear difference comparing to the rest of classes, indicating that their susceptibility to burn has increased over the 50-year study period. W (-0.03) and E (0.001) show insignificant differences with the first period while N-facing (-0.31) appear to be even less prone to fire. Flat surfaces continue to be avoided by fire at

lower degree, while having a much wider dispersion of samples, especially when comparing to the rest of the classes in both periods.

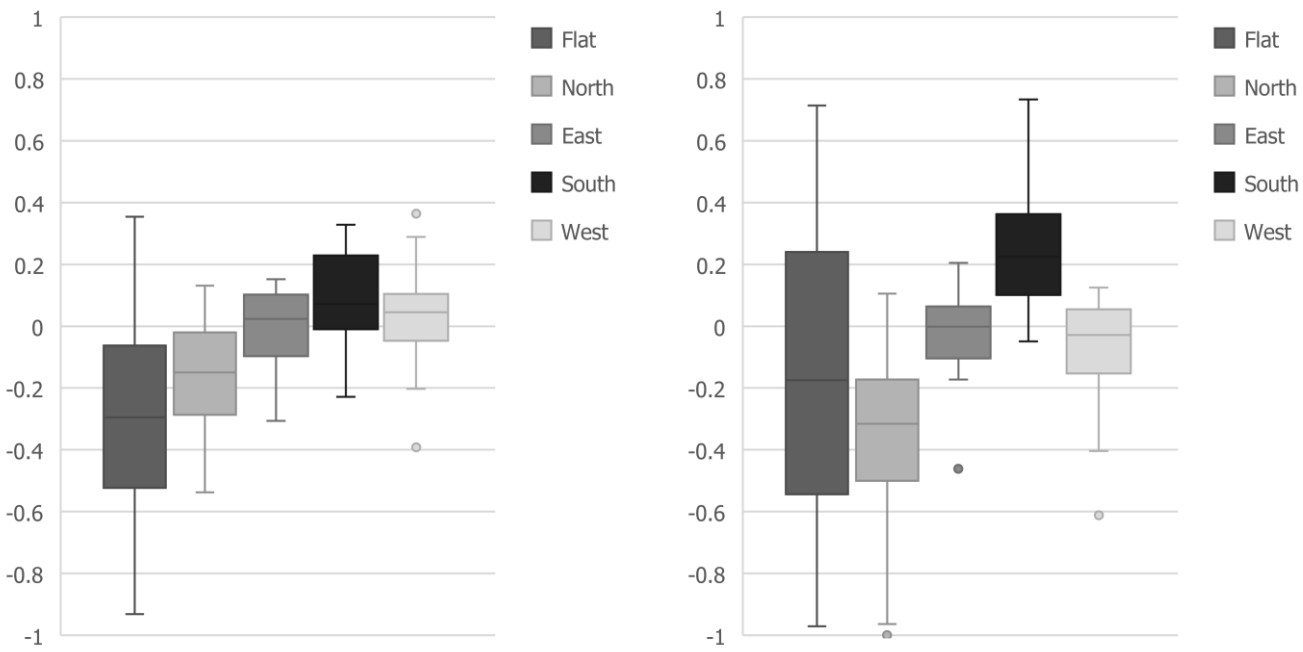


**Figure 10: Boxplot representing the distribution of Jacobs' index (ranging from -1 to +1) for 1970-1994(left) and 1995-2019(right) according to slope aspect. i) Median value (50th percentile): bar within the box, ii) first quartile (25th percentile): bottom part of the box, iii) third quartile (75th percentile): top part of the box. Whiskers represent observations outside the middle 50% and points represent outliers.**

**3.2.2 Slope inclination**

As for slope aspect, figure 11 shows fire selectivity for each of the two periods based on Jacobs' selectivity index according to slope inclination. During the first period, the lowest (≤10°) and the highest (>40°) slope inclination categories tend to be avoided by fire according to median fire selectivity (-0.28 and -0.2 respectively). Intermediate slope categories (10°-40°) have positive but close to zero median value and do not exhibit any solid fire selectivity pattern. In the second period, median fire

selectivity for mildest (≤10°) slopes shift from being avoided by fire to being weakly preferred (0.13). Steepest (>40°) slopes, located mainly in the eastern segment of the study area, shift closer to zero (-0.08). Low intermediate slopes (10°-20°) which account for a high percentage of BA in the western (Bouches-du-Rhône) and central (Var) parts of the study area shift from positive to negative selectivity (-0.09). Slopes within the 20°-30° range show a slight increase in median fire selectivity from 0.09 to 0.16. Steeper slopes (30°-40°) are marginally preferred by fire (0.1) while the steepest slopes (>40°) are still avoided

by fire but a lower degree.

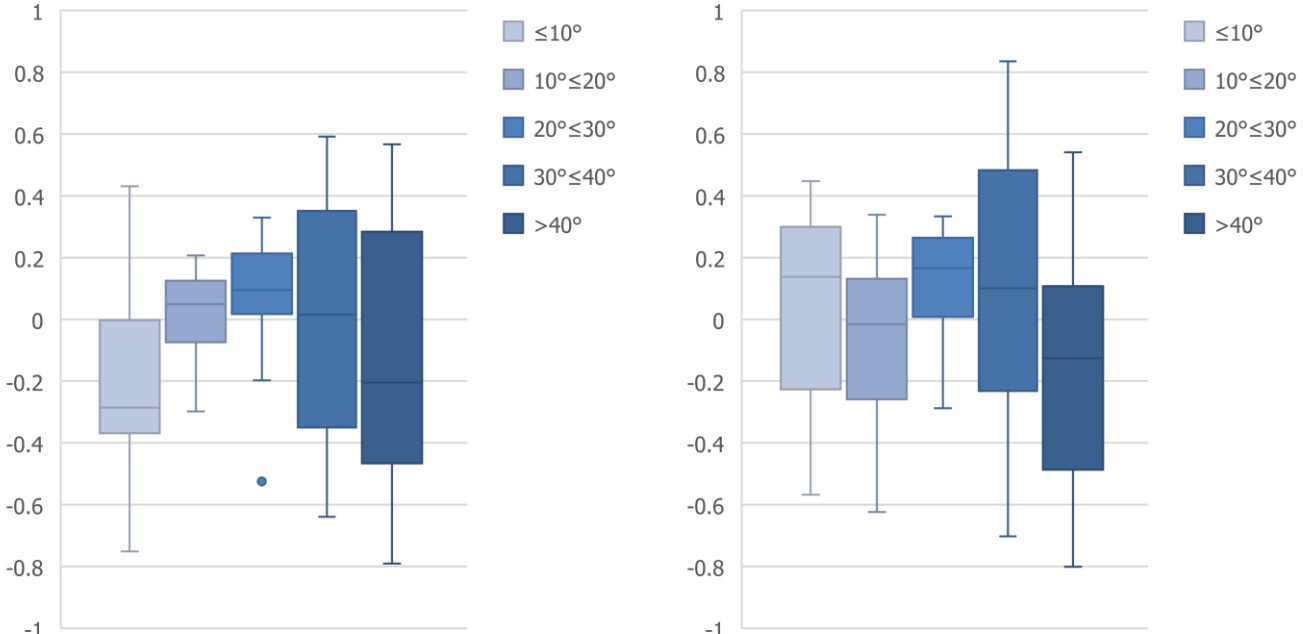

**Figure 11: Boxplot representing the distribution of Jacobs' index (ranging from -1 to +1) for 1970-1994(left) and 1995-2019(right) according to slope inclination. i) Median value (50th percentile): bar within the box, ii) first quartile (25th percentile): bottom part of the box, iii) third quartile (75th percentile): top part of the box. Whiskers represent observations outside the middle 50% and points represent outliers.**

## 3.3 Fire selectivity and vegetation type

Forested and semi natural vegetation is distributed between 5 categories of which grasslands and sclerophyllous vegetation have the lowest and the highest 50-year average covers, respectively: Broad-leaved Forest, 20.6 %; Coniferous Forest, 24.1 %: Mixed Forest, 19.2 %; Grasslands, 11.2 %, and Sclerophyllous, 24.9 %. Through the last 5 decades, mixed forest and broadleaves remained roughly the same in terms of coverage, whereas Conifers present a slight but decreasing trend. Sclerophyllous vegetation expanded in the study area (≈6 % increase), becoming the most common type in the last 3 decades. Finally, natural grasslands is by far the least common type and shrunk slightly (≈3,5 % decrease) over the course of the last 50 years.

**Table 3  Average and relative forested areas according to vegetation type between 1970 to 2019.**

| Type | Area (ha) | % |
|---|---|---|
| Broad-leaved Forest | 172,547 | 20.6 |
| Coniferous Forest | 201,262 | 24.1 |
| Mixed Forest | 160,973 | 19.2 |
| Natural Grassland | 93,322 | 11.2 |
| Sclerophyllous Vegetation | 208,057 | 24.9 |

| | | |
|---|---|---|
| Total | 836,161 | |

Same as the topographic factors, fire selectivity with regards to vegetation type is presented in figure 12. The rank from the most to the least fire prone vegetation types is: Sclerophyllous vegetation (0.28), Natural grasslands (0.26), Mixed forest (-0.27), Broad-leaved forest (-0.31) and Coniferous forest (-0.32). Even though the order changes slightly in the second period the effects of the fire suppression strategy on vegetation types are more evident than the topographic factors. Although with wider distribution, Sclerophyllous vegetation shows a nearly identical median value (0.26) and remains the most fire prone type. Fire selectivity for Natural grasslands has shifted and now appears neither preferred nor avoided by fire. Coniferous forests are still avoided by fire (-0.14), whereas both Mixed forest and Broad-leaved forest display significant alteration in median fire selectivity that is characterized by strong fire avoidance (-0.69 and -0.76 respectively).

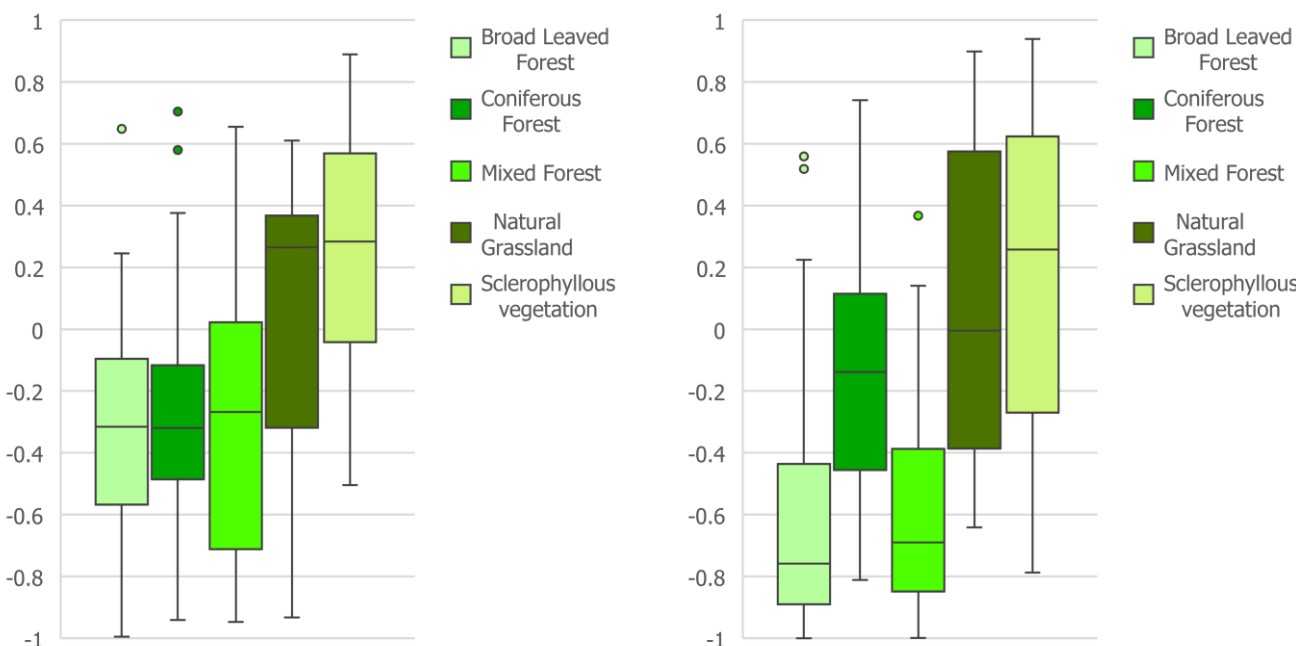

**Figure 12: Boxplot representing the distribution of Jacobs' index (ranging from -1 to +1) for 1970-1994(left) and 1995-2019(right) according to vegetation typee. i) Median value (50th percentile): bar within the box, ii) first quartile (25th percentile): bottom part of the box, iii) third quartile (75th percentile): top part of the box. Whiskers represent observations outside the middle 50% and points represent outliers.**

### 3.4 Geographically weighted regression

There is a considerable spatiotemporal variability in the strength of the correlation between the BA and environmental variables throughout the study area. Coefficient of determination $R^2$ values range from 0.00 to 0.68 (Slope inclination) depending on the variable (table 4). Along with the topographic factors, Sclerophyllous vegetation show overall the strongest fit in the relationship with BA. The rest of the vegetation types display a weak fit that remains similar in both periods.

**Table 4 Descriptive statistics of local R² per environmental factor for period 1 (1970-1994) and for period 2 (1995-2019).**

| | Slope Orientation | | Slope Inclination | | Sclerophyllous Vegetation | | Natural Grasslands | | Coniferous Forest | | Broad-leaved Forest | | Mixed Forest | |
|---|---|---|---|---|---|---|---|---|---|---|---|---|---|---|
| Period | P1 | P2 | P1 | P2 | P1 | P2 | P1 | P2 | P1 | P2 | P1 | P2 | P1 | P2 |
| Minimum | 0.00 | 0.00 | 0.00 | 0.00 | 0.00 | 0.00 | 0.00 | 0.02 | 0.01 | 0.00 | 0.00 | 0.00 | 0.00 | 0.00 |
| Maximum | 0.24 | 0.36 | 0.68 | 0.25 | 0.48 | 0.47 | 0.19 | 0.21 | 0.20 | 0.21 | 0.11 | 0.23 | 0.26 | 0.25 |
| Mean | 0.08 | 0.11 | 0.13 | 0.06 | 0.19 | 0.17 | 0.07 | 0.08 | 0.08 | 0.09 | 0.04 | 0.06 | 0.07 | 0.05 |
| Standard Deviation | 0.08 | 0.12 | 0.08 | 0.04 | 0.12 | 0.11 | 0.03 | 0.05 | 0.03 | 0.03 | 0.02 | 0.05 | 0.05 | 0.04 |

Figure 13 depicts local R² results of the application of GWR between percentage of BA and topographic factors. Overall, highest values are concentrated mainly in western and central parts (closer to the coastline) of the study area both slope aspect and inclination. The proportion of variance explained by slope orientation is slightly higher in the second period with several cells being in the highest class (0.25-0.35). Despite having a strong local fit in the first period, both distribution and variability changed drastically for slope inclination in the second period.

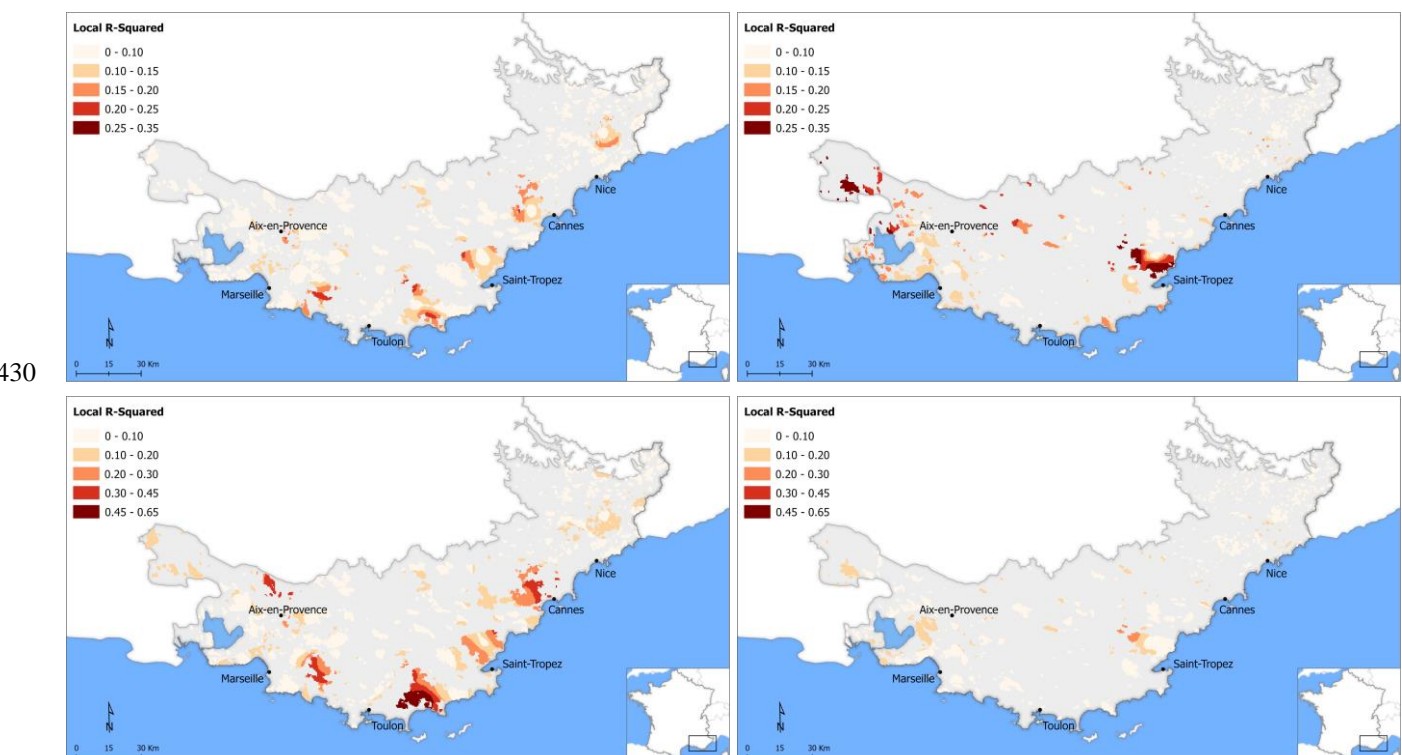


**Figure 13 Spatial distribution of local R² between burned area and slope aspect (top), slope inclination (bottom) for 1970-1994 (left) and 1995-2019 (right).**

Figure 14a and 14b display local $R^2$ results of the application of GWR between percentage of BA and percentage of each

vegetation type. Similar to topographic variables, Sclerophyllous vegetation exhibits the same spatial pattern of high $R^2$ values. A clear increase in local $R^2$ can be observed when moving towards the western part of the region, that is more evident in the first period. Low fits are found for both periods in the higher altitude areas, located mainly in north-eastern segments of the area. $R^2$ values for Natural grasslands display small differences both in terms of space and variance.

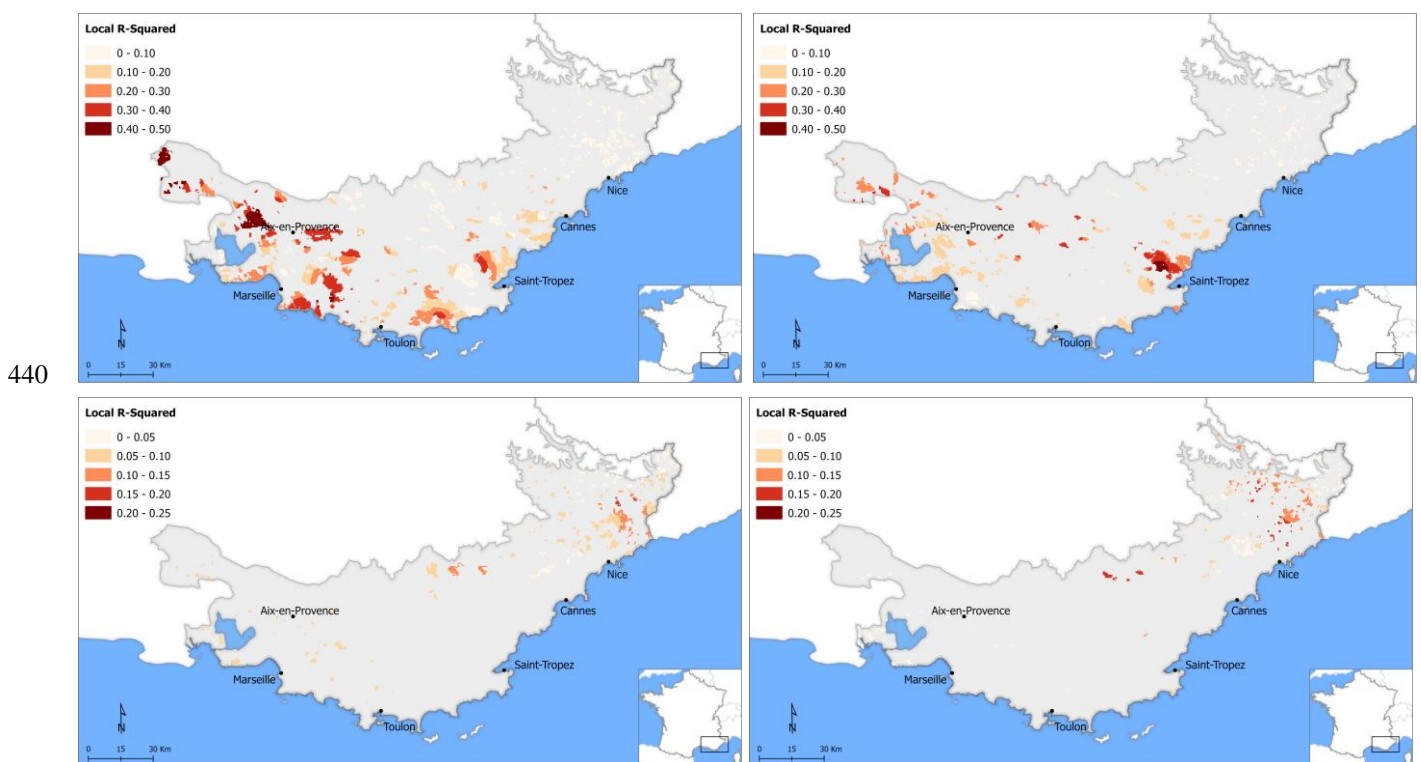


Figure 14a Spatial distribution of local $R^2$ between burned area and % cover of vegetation types (top: Sclerophyllous vegetation, bottom: Natural grasslands) for 1970-1994 (left) and 1995-2019 (right). Explanatory variables related to forest categories show very weak fit in the relationship with BA while also the general clustering patterns are quite different between

the periods.

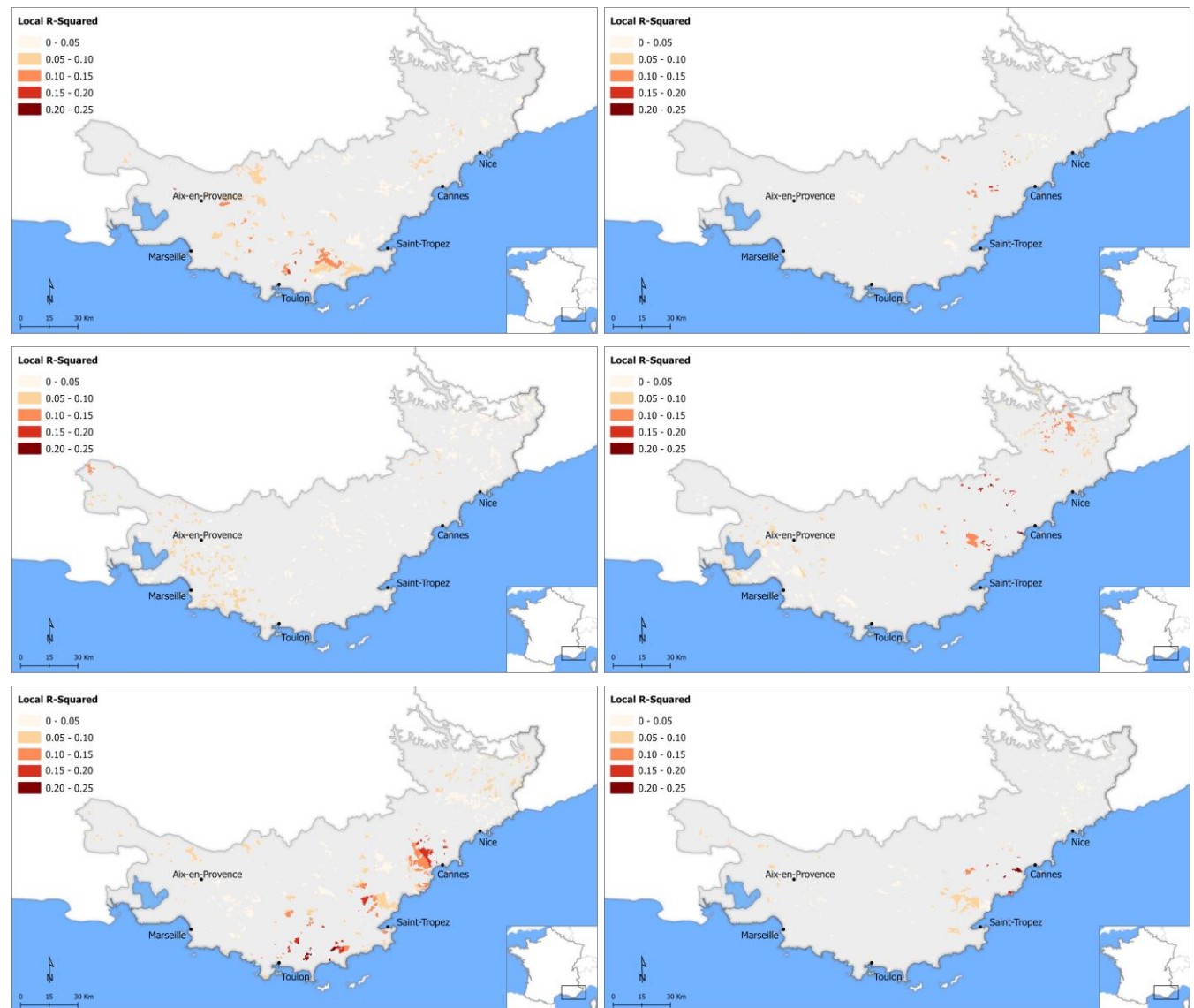

**Figure 14b Spatial distribution of local R2 between burned area and % cover of vegetation types (from top to bottom: Coniferous forest, Broad-leaved forest and Mixed forest) for 1970-1994 (left) and 1995-2019 (right).**

## 4 Discussion

### 4.1 Fire history

BA in south-eastern France has undergone substantial changes over the last 50 years. Clear declining trends characterize every decade after 1990 and especially in the 2010-2019 interval, where BA represent only 5.3 % (13,649 ha) of the total, which is a 68.9 % decrease compared to the previous decade (2000-2009). Furthermore, around half of the total BA (126,700 ha) was

recorded in 5 years: 1979, 1986, 1989, 1990 and 2003. Due to the catastrophic fires between 1986 and 1990, a new fire policy ("Vulcain") was established that came fully into effect in 1994 (Direction de la Sécurité Civile, 1994). This new strategy focused on aggressively suppressing fire ignitions under any weather conditions in order to avoid the propagation of fire to the extent where suppression would become both more difficult and more expensive. Although Fire Weather Index values were not calculated here for the 3 administrative departments, Fox et al. (2015) noted a general increase in summer temperatures between about 1980 and 2010, so the fire-fighting policy had a major impact on the decrease in total BA after 1994, with the exception of 2003. This year was the hottest/driest year on record in the Alpes-Maritimes since at least 1973 (Fox et al., 2015), and although its BA was greater than any other year since 1991, it was not exceptional compared to the 1979-1990 time interval. Nonetheless, it raised doubts regarding the sustainability of the rapid extinction strategy and its ability to reduce fire risk in the long term since resources are spread thinly over a greater number of ignitions (Curt and Frejaville, 2018).

### 4.1.1 Spatiotemporal analysis

The effect of the new firefighting strategy can also be viewed spatially; fire patches are less large and are distributed over smaller geographic proximities with one another, and fire recurrence is lower and clustered mostly near populated zones. Despite the size of the study area and the physical proximity of the burn scars, different factors contribute to determine fire patterns before and after the policy change. In the western part of the study zone, around Aix-en-Provence and Marseille, where low vegetation continuity affects fire propagation and size, the change in fire-fighting strategy had less effect since fires were already limited in size by vegetation patch sizes. Even though limited in terms of area, multiple clusters of significant increasing trends are spread in closer proximity to built-up areas in comparison to patches of decreasing trends. Generally, wind speed is a particularly important factor for BA (Duane et al., 2015; Fernandes et al., 2016) and in this part especially since it is the main driver of large fires (Ruffault and Mouillot, 2015, 2017). On the contrary,the central part of the study area, where most of the big scars are located, the new fire policy was able to effectively limit the propagation of fire over the continuous forest cover that defines the region. This region displays the largest clusters of decreasing BA and very limited areas with increasing BA which indicates that if trends remain similar over the next years, the area should possibly anticipate a further decrease in fire activity. Ganteaume and Barbero, (2019) provided evidence that large fires (>100 ha) are declining sharply in the central segment of the study area after the introduction of the fire management policy. In the eastern segment of the study area, which involves different environmental characteristics (higher elevation, milder winds, high fuel continuity), many small and dispersed patches are found, Fire shapes are not elongated by wind direction comparing to rest of the study area and were reduced in the low-altitude WUI near-coastal areas in the second period (1995-2019). Despite the higher fuel continuity and smaller effect of wind speed a similar pattern with the eastern section is present here; small but significant increasing trends appear closer to areas of increased human activity, which is known to affect fire ignition (Badia et al., 2011; Chas-Amil et al., 2013; Jiménez-Ruano et al., 2017; Lampin-Maillet et al., 2011).

## 4.2 Burned area and topography

### 4.2.1 Slope orientation

S-facing slopes have the greatest BA, burn more frequently (Mouillot et al., 2003) and are more exposed to forest fires than other slopes due to both environmental factors (greater insolation and evapotranspiration) and WUI characteristics since S-facing slopes in southern France have more houses and therefore more potential ignition sources (Fox et al., 2018). Generally S-facing (sum of SW, S, SE) slopes play an increasingly important role over time, which could be linked to a combination of hotter summers and an increasing number of human dwellings on these slopes as growth rates on S-facing slopes in the Alpes-

Maritimes were 4-5 times greater than on N-facing slopes in 1990-2012 (Fox et al., 2018). However, considering that the fire policy is contributing in weakening the of fire-weather relationship (Ruffault and Mouillot, 2015), human presence has potentially a larger influence in that increase.

### 4.2.2 Slope inclination

Slope inclination favors fire propagation directly through more efficient radiative heat transfer (Rothermel, 1983) and increases

the rate of spread and fire intensity (Csontos and Cseresnyés, 2015; Capra et al., 2018). In addition, slope inclination influences fire ignition and suppression indirectly through accessibility, solar radiation variations, fuel moisture, and fuel density which in turn influence flammability (Holden et al., 2009). In this study, intermediate slope inclinations were most susceptible to burn for several reasons: radiative heat transfer is more efficient, they are too steep for dense human occupation but well-suited to isolated or diffuse housing, and accessibility for suppression is difficult. Normally, lowest inclinations are avoided by fire

since not only is radiative heat transfer less efficient on these slopes but flat areas are more densely inhabited and easily accessible, so the lower fire preference probably depends as much or more on early suppression as on physical processes. However, in the second period, lowest inclinations do not display the same fire avoidance. In addition, the results show a shift in BA from steeper to less steep slopes over time, and this suggests that the rapid suppression strategy put into place in the early 1990s reduced the propagation of fire from lower slopes where ignitions are expected to be greater to steeper wildland

areas. Fires were contained more quickly and escaped to steeper wildland areas less frequently. A similar temporal shift from steeper to flatter slopes is observed for fire frequency in west-central Spain (Viedma et al., 2018). Considering the negative relationship between slope inclination and human-caused fire ignitions found in Narayanaraj and Wimberly (2012), this can potentially be linked to the increasing human presence in flatter areas but also to more effective and rapid suppression.

Due to the complex interactions between physical processes, human activities and topography, topographic roughness is

frequently recognized as a significant but secondary predictor of fire occurrence after WUI characteristics in Euro-Mediterranean zone (Ganteaume and Jappiot, 2013; Nunes, 2012).

## 4.3 Burned area and vegetation type

The role of vegetation in fire frequency and BA patches located in the Bouches-du-Rhône and Var departments was studied by Curt et al., (2013). Their case study reflects patterns observed here at a larger scale, namely that vegetation flammability is secondary to landscape organization. Large open patches of continuous fuel, as are found in the Var department, favor larger fires with longer return intervals than the small patchy wildland distribution in the Bouches-du-Rhône (Ganteaume and Barbero, 2019). Burned vegetation patterns observed here highlight the frequently cited role of sclerophyllous vegetation (shrubland) (Ganteaume and Jappiot, 2013; Moreira et al., 2011; Oliveira et al., 2014a; Tessler et al., 2016). Shrublands both favor fire propagation in dry conditions (Baeza et al., 2002) and result from recurrent fires (Tessler et al., 2016). As Mermoz et al., (2005) suggested, the fire proneness of sclerophyllous vegetation is connected to its ability to regenerate faster and allow for quicker fuel accumulation, which also applies in our case since sclerophyllous vegetation is the type burning the most while also growing within the region. Additionally, in Mediterranean environments, large fires tend to occur in landscapes with dense shrublands (Moreira et al., 2011; Ruffault and Mouillot, 2017) and that is also the case here, since in all 5 years with the highest BA, sclerophyllous vegetation was the one with the greatest burned proportion. In a context where initial suppression is crucial to fire extinction, bushlands may resist early suppression better than other covers where initial propagation is perhaps slower. Moreover, firefighting assets appear to prioritize other types of vegetation during fire suppresion since fire selectivity remains unchanged for bushlands, possibly due to the low cost of restoration (Oehler et al., 2012).

As other studies have concluded (Oliveira et al., 2014a), natural grasslands display a high fire susceptibility. Prior to the change the in the fire policy, grasslands are over-represented in BA and this may be due to faster initial propagation or accessibility issues, as for example in certain mid to high-altitude areas over the eastern section of the study area, where burned clusters of this vegetation type are found. However, fire does not favor equally the specific type in the second period potentially due to improved fire suppression methods.

On the other hand, conifers have low BA proportion even in years when big fires occurred. Coniferous forest is a recognized pyrophyllic cover, but it is the least fire prone type in our results, possibly because most of the coniferous area is in high-altitude zones where there are fewer fire ignitions. This also confirms the importance of landscape organization described by Curt et al. (2013).

## 5 Conclusion

In this study, results provide a coherent picture of interactions between a long temporal fire geodatabase and environmental characteristics through the scope of changes in firefighting strategies. In areas of relatively high artificial densities, as in the Bouches-du-Rhone, BA patches are limited in size by fuel continuity, despite recurrent strong "Mistral" winds. Areas with large continuous vegetation and diffuse human occupation are most propitious to large frequent fires, as in the Var department. Remote hinterlands with extensive continuous vegetation have fewer fires, as in the Alpes-Maritimes due to a lack of ignition

sources. Decline of large patches of fire is clear after the policy but new clusters of high fire recurrence appear closer to areas with increased human activity.

S-facing aspects have an increasingly bigger impact over time, which may be linked to both environmental and increased human presence on those slopes. Slope inclination remains important throughout the 50-year study period during which intermediate inclinations (20-30°) are more susceptible to burn; this is probably due to both physical processes (radiative heat transfer) and human considerations (accessibility…), though these were not quantified here. The general shift in BA from steeper to gentler slopes over time may reflect the impact of early suppression as described above.

Over half of the total BA in the last 50 years concerned sclerophyllous vegetation, thus confirming its strong association with high fire susceptibility and recurrence. Considering that sclerophyllous vegetation regenerates and expands faster than the rest of the vegetation types in the region, this may lead to an increase in fire risk in the future. Natural grasslands, even though they cover limited area and decline with time, were also preferred by fire prior to the change in fire management policy.

Further ongoing exploitation of the fire GIS database in conjunction with WUI characteristics will likely further improve our
understanding on the driving forces of BA and the impacts of fire-fighting strategies in the region.

**Author contribution**

CB established the fire geodatabase, carried out data processing, analyses, visualization and wrote the initial draft. DF performed the land cover modeling, contributed to the interpretation of the results, wrote, and reviewed the manuscript. EB provided expertise for data analyses and reviewed the manuscript.

**Competing interests**

The authors declare that they have no conflict of interest.

**Acknowledgements**

This work has been supported by the French government, through the 3IA Côte d'Azur Investments and the Future project managed by the National Research Agency (ANR) with the reference number ANR-19-P3IA-0002.The authors also gratefully
acknowledge the contribution of Université Côte d'Azur Academy-3.

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
