# Peer review of "Environmental Factors Affecting Wildfire Burned Area In South-Eastern France, 1970-2019"

_Natural Hazards and Earth System Sciences, 2021_

## Author Comment (AC1)

Dear reviewers,

We are grateful for the constructive comments that help improve our manuscript substantially. In our response, we would like to describe how we intend to address the suggestions and the matters that were brought up.

We believe that the scientific robustness of the manuscript will be significantly enhanced in the revised version, and we hope that the reviewers approve the changes that will be implemented. Overall, our focus will shift towards a more quantitative and less qualitative approach and will provide more references to other significant studies. Some superficial corrections have already been brought to the manuscript, but more fundamental issues of data treatment remain to be done.

Our response is divided in two major sections. The first part refers to the methodological issues that were raised by the reviewers, while the second one addresses specific comments from each reviewer individually. Since the methodological concerns that were brought up are very similar from both reviewers, we included a common response to that matter.

**1. Methodological Issues**

As suggested by both reviewers, we agree that the methodological framework of the manuscript can be greatly improved, and we believe that the proposed approaches are appropriate and very useful additions.

**1.1 Selection Indices**

The specific approach supports impeccably our goals and helps us mitigate the limitations of the specific manuscript. An implementation of a Selection Index will be included in the revised version, as in other similar studies that were suggested (Moreira et al. 2009; Moreno et al. 2011 and Barros et al. 2014), to tackle the subject of fire selectivity and improve the statistical state of the manuscript.

**1.2 Trend Analysis**

As it's clear that the change of fire suppression policy had an impact on burned area, we aim to briefly touch on the specific subject. Following the suggestion of reviewer #1, we will implement an analysis of the fire trends over our study area using the Man-Kendall trend analysis.

**1.3 Spatial Patterns**

The spatial patterns of burned areas is one of the basic aspects that we intended to address in the specific manuscript. As such a quantitative assessment of burned area patterns will be included in the revised version as pointed out by the reviewers. The fire geodatabase that was utilized in our study, can provide us with detailed perimeters of the burned area and we aim to fully take advantage of that.

**1.4 Size Patterns**

Even though this is certainly an interesting subject, we don't plan to consider it for the current manuscript as we aim to explore fire size in a separate study.

**1.5 Geographically weighted regression**

We intend to build our analysis using a Geographically weighted regression. Additional variables such as vegetation connectivity, wind speed, elevation and others are being explored and will be added in the list of the current environmental factors. The importance of those factors will be assessed using the specific approach.

**2. Specific comments**

Please find below our responses to your specific comments and suggestions. Reviewers' comments are in black italic font and our answers are in blue bold font and are placed right below each comment.

**2.1    RC1 (Anonymous Referee #1)**

*In introduction, other several references could be included to support many sentences. Overall, the objective of this paper is to assess temporal changes in BA spatial patterns, but there is not any reference about this topic.*

*Please see: Viedma et al. 2018 although it is related to fire number (fire frequency). Viedma, O., Urbieta, I. R., & Moreno, J. M. (2018). Wildfires and the role of their drivers are changing over time in a large rural area of west-central Spain. Scientific reports, 8(1), 1-13.*
*For example, to support this sentence: Among the environmental characteristics, several studies provide evidence of spatial patterns relating topography to forest fire probability (Dickson et al., 2006; Padilla and Vega-García, 2011)" Please, read and include this paper: Viedma, O., Urbieta, I. R., & Moreno, J. M. (2018). Wildfires and the role of their drivers are changing over time in a large rural area of west- central Spain. Scientific reports, 8(1), 1-13."*

**We thank you for bringing to our attention the specific paper, it's clear that there are many aspects that are useful in our paper.**

*To support this sentence: "Csontos and Cseresnyés (2015) observed an exponential velocity increase in upslope fire spread with the increase in slope inclination whereas downslope fire spread velocity was unaffected by slope angle and was similar to rates detected on flat terrain"*

*Please, read these papers (although they are devoted to fire severity, they explain how upslope fire spread caused greater Rate of Spread, and consequently higher severity):*
*Viedma, O., Quesada, J., Torres, I., De Santis, A., & Moreno, J. M. (2015). Fire severity in a large fire in a Pinus pinaster forest is highly predictable from burning conditions, stand structure, and topography. Ecosystems, 18(2), 237-250.*
*Viedma, O., Chico, F., Fernández, J. J., Madrigal, C., Safford, H. D., & Moreno, J. M. (2020). Disentangling the role of prefire vegetation vs. burning conditions on fire severity in a large forest fire in SE Spain. Remote Sensing of Environment, 247, 111891.*

**The suggested papers are cited in the revised version of the manuscript and will help us support the specific claim.**

*To support this sentence: "…and the probability of large fires in landscapes with dense shrublands is greater than in forested ecosystems in the Mediterranean basin (Moreira et al., 2011; Ruffault and Mouillot, 2017)."*

*Please, read this paper: Urbieta, I. R., Franquesa, M., Viedma, O., & Moreno, J. M. (2019). Fire activity and burned forest lands decreased during the last three decades in Spain. Annals of Forest Science, 76(3), 1-13. Here, authors showed that treeless areas tend to burn more than treed areas in Spain during the last decades.*

*To support or enlarge this sentence with the trends during the last decades: "…broadleaved forests are usually less prone to burning than coniferous species which present a greater fire hazard (Moreira et al., 2009; Oliveira et al., 2014)." Please, see Urbieta et al. 2019: In Spain oak forests are burning more than conifers in the last decades.*

**We are thankful for bringing to our attention this recent paper. It's a valuable addition to our list of references.**

*On the other hand, there is a great confusion with the cell size of the grid to extract the frequency data. For example, in Lines 124: "A 500x500 m grid was created and overlaid on the study area in order to measure the percentage of the area that was burned inside every 25 m cell for each year". Later, author say that each cell is 25 ha but a 25 x 25 m cell is 625 m2. Sorry, but I don't understand anything. Please, clarify this.*

**There is indeed a typo with the units of area in line 124, which is now corrected (25 ha).**

*In addition, it is said that "…BA by vegetation type used the CLC layer closest to the BA data"*

**The specific sentence has been rephrased to improve its clarity.**

**"The fire geodatabase was split into seven-time intervals and subsequently matched with the CLC layer that is chronologically closest to the equivalent fire period (see Table 1)."**

*Please, be careful because the dates of the LULC maps were not before of several forest fires, and we expect that you have checked that the LULC represented by the maps always indicated prefire vegetation.*

**We thank the reviewer as this comment gives us the opportunity to make our work clearer. In the case that an area was classified as "334 - Burnt areas" in a given CLC dataset, a different CLC layer (preceding the fires) was used instead to replace burned surfaces with the equivalent pre-fire vegetation.**

*In fig. 5 and later over the paper, you use the term "fuel type". Please, change it by forest type or vegetation type or even Land cover type; because you are not working with fuel types, but only with land covers.*

**As it was correctly pointed a different term would be more appropriate. We decided that the term "vegetation type" would be the instead.**

*Please, improve the quality of fig. 6 and others done in Excel. Remove internal lines of the plot and be careful with the borders of the figure. Letters in black better than gray.*

**The figures have been adjusted based on your feedback. Internal lines have been removed and letters are now in black.**

*This paragraph must be in discussion section, not in results: "Overall, the patterns described here are coherent with known interactions between fire ignition, vegetation continuity, and wind speed: fire ignition occurs most frequently in proximity to human activities (Badia et al.,*

*2011; Chas-Amil et al., 2013; Jiménez-Ruano et al., 2017; Lampin-Maillet et al., 2011) and BA depends on fuel continuity and wind speed (Dueane et al., 255 2015; Fernandes et al., 2016). "*

**We agree with the reviewer that the specific paragraph should be in the discussion section. It's now under section "4.1. Fire History".**

*This sentence reflects one of the limitations of this paper: Line 265: "However, on the northern shore of the Mediterranean, there are generally more S-facing (sum of SW, S, SE) than N-facing (sum of NW, N, NE) slopes, and BA distribution may therefore be a simple reflection of area rather than susceptibility to burn. "…"In order to compensate for this, BA is plotted as a percentage of the burned forested slopes As you say, this is a limitation and other type of statistical analysis should be carried out as the resource Selection Index: See this paper: Moreno, J. M., Viedma, O., Zavala, G., & Luna, B. (2011). Landscape variables influencing forest fires in central Spain. International Journal of Wildland Fire, 20(5), 678-689.*

**The specific topic has been addressed in the section above**

*There is confusion in the figure numeration: It is not figure 9 but figure 8. The same with figure 10 in line 261 (it is figure 8) To support this sentence and make comments in discussion section:*

**Indeed, there is a mistake with the figure numeration. The correction has been addressed.**

*Below 30°, there are no clear differences between slope inclination categories. Above 30°, the percentage BA drops abruptly. Temporal fluctuations of the distributions show a general shift from high inclination slopes (40° or greater) to lower inclination slopes (≤20°)." Results are like those found in Viedma et al. 2018: a shift of fire frequency to flatter areas. See: Viedma, O., Urbieta, I. R., & Moreno, J. M. (2018). Wildfires and the role of their drivers are changing over time in a large rural area of west-central Spain. Scientific reports, 8(1), 1-13.*

**We thank the reviewer for this suggestion as it complements our results. The suggested literature is added in section "4.2.2 Slope inclination".**

*Please reconsider to change the figure caption of figures 12 and 13. I propose these:*

*Fig. 12. Percentage of burned vegetation according to the area of vegetation types by decade*

*Fig. 13. Percentage of burned vegetation according to the total burned area by decade*

**Figures have been renamed to:**

**Fig. 12: Distribution of burned vegetation according to the area of vegetation type by decade (the sum of each decade equals 100%).**

**Fig. 13: Percentage of burned vegetation according to the total burned area by decade.**

**2.2 RC2 (Anonymous Referee #2)**

*(i) Perhaps, my main concern regards the analyses whose results are shown in Fig, 8,9,10, 11, 12, and 13) and that do not seem appropriate for the objectives raised in this paper. Indeed, despite an overall agreement of your results with the literature, there is no guarantee that the conclusions drawn from your current analyses are not affected by other drivers of fire spread, such as fuel connectivity, weather conditions, and many other possible interactions including those that might occur between your studied factors. The presentation of results is very convoluted (see for instance the section from L261 to L279 that is very difficult to follow) and no framework currently rigorously demonstrates that LULC influence is significant from a statistical point of view. In light of these remarks, it seems essential to rethink the analytical framework to ensure that these three points are correctly addressed by developing a more appropriate analytical framework that takes into account (or at least minimizes) these interactions and presents the results in a more relevant manner. One interesting possibility would be to determine whether the observed values of fire selectivity are significantly different from those that would be observed if fires were to occur at random locations in the landscape. I would advise you to look at the methodologies shown in Moreno et al. (2011) and Barros et al. (2014) that seem relevant because they assess fire selectivity through a null-based model that is independent of spatial relationships or any bias of fire data. Note that other approaches might also be relevant. Besides I found that the approach developed in Fig 7 and the conclusions drawn from these analyses were highly descriptive and speculative (see L224-L255). No analyses on the spatial pattern are provided, simply a description of the maps. These analyses should therefore also be improved to strengthen the results and their interpretation.*

**We agree with your concerns regarding the analytical framework of the manuscript and as suggested by both reviewers, this is an important drawback and is currently being dealt with. The matter has been addressed in the "1. Methodological Issues" section of our response.**

*(ii) I was surprised by the results in Fig 6 showing no trend in fire numbers, as it differs from the decreasing trend generally reported for this area see for instance Fig. 2 in Curt and Fréjaville, 2018). A possible explanation for such a discrepancy is that your database could be biased by a non-constant reporting of fires over time, which in turn might affect your results and conclusions. What I suspect is that a higher proportion of fires are now reported compared to what it was in the early 1970's especially for the smallest fires. To reduce the potential biases induced by this inconsistency, one solution would be to compare your dataset against the french promethee fire database for fires > 1ha (that is currently considered to be a relevant and robust fire size threshold to study fire ignitions in France, Pimont et al. 2021) in order to determine the fire size threshold above which you consider your database is not affected by an evolution of detection /reporting over time. Furthermore, I think that this manuscript would really benefit from including, somehow, a size factor in the analytical framework. Indeed, it would be relevant to test whether under severe fire weather, fires are expected to become larger and less dependent on land cover, which is generally reported in southern Europe.*

**Indeed, in Curt and Fréjaville (2018) there is a clear decreasing trend with regards to the number of fires in the region but there are some differences to keep in mind between the two manuscripts such as: A) The study area is significantly larger and includes data from all departments (15 in total, comparing to 3 in our manuscript) available on Promethee. Furthermore, the study area in our manuscript corresponds**

to the departments with the greatest BA in France (after Corsica) in comparison with the Curt and Fréjaville (2018) which includes departments that have considerably lower fire activity. B) The period under study ranges from 1975 to 2014, while in our study extends to 2019. In that period (2015 to 2019) the number of fires increased, and especially in 2017 when the highest number of fires in the last 17 years was recorded.

Figure 1 shows the burned area and the number of fires larger than 1 ha (only for our study area) using the same database as in the manuscript under review, and Figure 2 shows the equivalent metrics based on the Promethee database. The specific threshold (>1 ha) does not seem to impact in a meaningful way the difference between the fire databases. There is evidently a big disparity in the recorded number of fires that appears to slightly decrease with time, as suspected. The trend in fire numbers is significant in figure 2 while in figure 1 the trend is gentle but still present. However, as we only deal with burned area here this is not really a problem since the resemblance between the two databases is high (correlation coefficient 0.522 for number of fires and 0.996 for burned area). The contrast in fire numbers may be attributed to the fact that ONF database provides the exact fire perimeter (which needs to be mapped using the available means of each period), in comparison to the Promethee database where an approximate area of a fire is sufficient.

Regarding the inclusion of fire size in the current analytical framework is certainly interesting and is planned to be explored in a separate research paper, potentially utilizing the Promethee database as it appears to be less biased with regards to fire records.

[Figure]

*Figure 1 Annual burned area and number of fires (>1 ha) from 1973 to 2020 (ONF database)*

[Figure]

*Figure 2 Annual burned area and number of fires (>1 ha) from 1973 to 2020 (Promethee database)*

*(iii) I found that references to previous studies were not enough detailed, sometimes vague, or even missing and my opinion is that this manuscript could be greatly improved on this matter. As mentioned in one of my previous comments, several important papers, whose vast majority address fire regimes over the Iberian Peninsula, investigate the impact of LULC on fires but were not cited in this manuscript, including for instance Carmo et al. (2011), Bajocco et al. (2008), Moreno et al. (2011), Nunes et al. (2005), Koutsias et al. (2012) among many others. Furthermore, like any other area, the French Mediterranean region has its own peculiar context and history regarding fire regimes, whose description and analyses could also be improved. For instance, in France, the papers from Fréjaville and Curt (2016), Ruffault and Mouillot (2015), Ruffault et al. (2016), and Evin et al. (2018) studied the consequences of the introduction of this new fire policy on different metrics. Some of their results and discussions might provide relevant results for your discussion and the building of your hypotheses. There are a few papers also that have explored the drivers of spatial fire weather and or fire hazard in Southern France that might help, including the works from Ruffault et al. (2017) and Pimont et al. (2021). Note that this is by no means a list of papers that need to be cited but rather some references that the authors you might find useful to improve the quality of your manuscript.*

**We greatly appreciate the list of literature that is suggested here, as well as the specific reasoning and motivation behind each recommendation. Undoubtedly, the articles exploring the impact of LULC on similar environments as well as the ones dealing specifically with the effects of the new fire policy in our context, will improve the quality of the manuscript by a great margin.**

**References**

Barros, A. M. G., & Pereira, J. M. C. (2014). Wildfire selectivity for land cover type: Does size matter? PLoS ONE, 9(1). https://doi.org/10.1371/journal.pone.0084760

Curt, T., & Frejaville, T. (2018). Wildfire Policy in Mediterranean France: How Far is it Efficient and Sustainable? Risk Analysis, 38(3), 472–488. https://doi.org/10.1111/risa.12855

Moreno, J. M., Viedma, O., Zavala, G., & Luna, B. (2011). Landscape variables influencing forest fires in central Spain. International Journal of Wildland Fire, 20(5), 678–689. https://doi.org/10.1071/WF10005

Moreira, F., Vaz, P., Catry, F., & Silva, J. S. (2009). Regional variations in wildfire susceptibility of land-cover types in Portugal: implications for landscape management to minimize fire hazard. *International Journal of Wildland Fire*, *18*(5). https://doi.org/10.1071/WF07098

---

## Author Response (AR2)

Dear reviewers,

We would like to express our gratitude once again for your comments and suggestions that have further improved the manuscript.

Below are our responses to the second iteration of reviews. Reviewers' comments are in black italic font and our answers are in blue bold font and are placed right below each comment.

**RC1 (Anonymous Referee #1)**

*Dear authors:*

*You have answered correctly the questions raised in the peer-review process; and consequently, the manuscript has improved. Nevertheless, there are some minor issues that must be accomplished before publication:*

*There are several typos and English language missuses in the paper that need to be revised. For example, vegetation types are in upper letters; and the description of the equation in line 290 must be consistent (for example, I or i (for location); among others.*

**Thank you for bringing up these issues. The text has been read and corrected. All categorical variables were given capital letters in order to be consistent.**

**RC2 (Anonymous Referee #2)**

*It is my second review of this manuscript. I would like to thank the authors for their relevant and detailed responses to my last comments. The modifications made by the authors improved the quality and readability of the manuscript. However, the manuscript requires a significant rewriting before I feel comfortable recommending it for publication. I detail below the points that I would like to be addressed by the authors.*

*- The introduction of a selectivity index (Jacob's index) undoubtedly strengthened the manuscript results compared to the previous version. But as applied in the present study, this approach also has two main drawbacks that should be either solved (by refining analyses) or (at the very least) discussed in a dedicated section. First, a working hypothesis of the authors is that the study is "relatively small",and they assume that "the available area to burn is defined as the total forested area in the region" (L255). I disagree with this assumption because it implies that all other fire drivers (fire weather, vegetation structure for a given class, land use) should be considered constant. I was under the impression that this is not the case in southern France, where climate and land use are spatially heterogeneous and characterized by steep gradients. It also contradicts the author's conclusion regarding the effect of coniferous type on BA (L583-584). The largest fires account for much of BA and generally occur under severe fire weather (under which bottom factors play a weaker role than in milder fire weather). I, therefore, expect the effect of land-cover to be underestimated. Both biases (and others not discussed here) should be deeply addressed in a dedicated section. I encourage the authors to add a section in the discussion.*

**Thank you for your detailed reasoning behind this point. Even though there is no universal rule defining the extent of the available area for a fire to burn, we decided to recalculate Jacob's index in a manner similar to Barros and Pereira, 2014 and Nunes et al., 2005, thus making the results comparable to those of similar studies. For each fire the available area to burn equals to twice the amount of area burnt.**

**a buffer zone around each fire perimeter that equals**

*- Despite the efforts made by the authors to strengthen the analyses of the spatial trends in fire activity in their study area, I remain unconvinced by the relevancy of this analysis (Figs 8 and 9). I can even recognize individual fires in these maps and I do not see how it could represent any general trends given the stochasticity in fire shapes. Perhaps, decreasing the spatial scale would decrease the impact of fire stochasticity. But, even then, I very much doubt that it could bring useful information. My advice here is to remove these results from the manuscript.*

**We thank you for your suggestion on this point. With the introduction of the Contextual Mann Kendal method in the revised version of the paper, the specific figures were meant to give an overview of the burned areas between the two periods and not as a method to analyze the spatial trends, since there is much more solid and robust method for that matter. However, by your comment, we realize that this is not clear especially since these figures are in the "Spatio-temporal analysis" section. We decided to keep the figures but move them to a different section ("Fire History").**

*- The number of fires recorded in the author's database is inconsistent with that of the national database (L142-143). Reporting these results without a detailed explanation of these differences might create some confusion among the readers not perfectly aware of the French fire context. A straightforward solution would be to remove fire numbers from the manuscript and reformat Fig 7.*

**We agree with your concerns, it is indeed simpler to remove the number of fires from the figure to avoid potential confusion with fire numbers in the Promethee database. The annual number of fires has been removed from the figure:**

[Figure]

*- There is a general tendency for the authors to overinterpret their results and focus their discussion on issues not directly addressed by their analytical framework nor related to their findings. There are numerous examples throughout the manuscript and I did not make the effort to note them all here. Perhaps, the conclusion of the manuscript is symptomatic of this tendency of overinterpretation of the results and discussing issues that are not directly related to the results shown in the present study. For instance, the authors said, that in "Bouches de*

*Rhone [..] BA patches are limited in size by fuel continuity, despite recurrent strong Mistral winds" while none of this driver has been explicitly investigated here. The following sentences are of the same type. I therefore strongly encourage the authors to carefully revise their manuscript and avoid, as much as possible, the overinterpretation of their results. In the case of the conclusion, that would mean to consider only the role of land-cover type and topography on BA in southern France.*

**We thank you for communicating the specific view. We have simplified the discussion and revised the text to avoid the problem of overinterpretation as you suggest.**

**References**

Barros, A. M. G., & Pereira, J. M. C. (2014). Wildfire selectivity for land cover type: Does size matter? PLoS ONE, 9(1). https://doi.org/10.1371/journal.pone.0084760

Nunes, M. C. S., Vasconcelos, M. J., Pereira, J. M. C., Dasgupta, N., Alldredge, R. J. and Rego, F. C.: Land Cover Type and Fire in Portugal: Do Fires Burn Land Cover Selectively?, Landsc. Ecol., 20(6), 661–673, doi:10.1007/s10980-005-0070-8, 2005.